# Lecture Notes on
# Normalizing Flows for Lattice Quantum Field Theories

**Miranda C. N. Cheng[a,b,c,1], Niki Stratikopoulou[b]**

**a** Korteweg-de Vries Institute for Mathematics, Amsterdam, the Netherlands
**b** Institute of Physics, University of Amsterdam, Amsterdam, the Netherlands
**c** Institute for Mathematics, Academia Sinica, Taiwan

## Abstract

Numerical simulations of quantum field theories on lattices serve as a fundamental tool for studying the non-perturbative regime of the theories, where analytic tools often fall short. Challenges arise when one takes the continuum limit or as the system approaches a critical point, especially in the presence of non-trivial topological structures in the theory. Rapid recent advances in machine learning provide a promising avenue for progress in this area. These lecture notes aim to give a brief account of lattice field theories, normalizing flows, and how the latter can be applied to study the former. The notes are based on the lectures given by the first author in various recent research schools.

---

[1]On leave from CNRS, France.

# 1   Introduction

Quantum field theories form a cornerstone of modern theoretical physics, an indispensable tool for understanding physical systems from small to large scales [1, 2]. At the same time, it is notoriously difficult to make them mathematically rigorous. Moreover, for many interesting theories, such as those that are strongly interacting, analytic computational methods often fall short. Lattice field theories constitute the only first-principle way of overcoming these difficulties, by making the problem finite-dimensional through a straightforward discretization of the space time. As one takes the continuum limit or as the system approaches a critical point, however, computations in lattice field theories become increasingly challenging. The challenge is exacerbated by the presence of non-trivial topological structures in the physical theory, such as the standard model for particle physics, due to the phenomenon of topological freezing. As the demand for precision in simulations increases, a more detailed understanding of the topological features, such as instantons and their associated susceptibilities, becomes crucial. Given the rapid advances of generative AI, opportunities arise to leverage these new tools to make progress in this physics problem of fundamental importance. In particular, a specific method in generative AI, called the normalizing flows, has been applied to the simulation of lattice field theories.

This lecture note aims to give a brief account of lattice field theories and gauge theories in particular, normalizing flows, and how the latter can be applied to study the former. We assume basic background knowledge in the physics of quantum field theories and some rudimentary understanding of basic machine learning concepts.

# 2   The World on a Lattice

In this chapter, we will review the basics of lattice field theories. In §2.1 we introduce the definitions in lattice field theories, and in §2.2 we introduce Monte Carlo sampling in the context of lattice field theories. In §2.3, we discuss the challenges such Monte Carlo sampling methods encounter, particularly in systems with non-trivial topological structures (§2.4). In the last subsection, we discuss the $\mathbb{C}P^{N-1}$ model as a simple example of a field theory with non-trivial topological features.

For concreteness, we first consider a simple example of a Euclidean scalar field theory. Consid-

ering the action of a scalar $\phi^4$ theory in $D$ dimensions,

$$S_{\text{cont.}}[\phi] = \int d^D x \left( \frac{1}{2} \partial_\mu \phi \partial^\mu \phi + \frac{1}{2} m^2 \phi^2 + \frac{\lambda}{4!} \phi^4 \right), \quad \phi : \mathbb{R}^D \to \mathbb{R}, \tag{1}$$

one replaces the space-time continuum with a $D$-dimensional grid, and restrict each of the field variables to be located at the vertices of a $D$-dimensional lattice with lattice spacing related to the UV cutoff scale via $a = \frac{1}{\Lambda_{UV}}$. To be concrete, we consider a $D$-dimensional periodic lattice of length $L$

$$V_L = \left\{ x : x \in \sum_{\hat\mu} a n_\mu \hat\mu, n_\mu \in \mathbb{Z}/L\mathbb{Z} \right\}, \tag{2}$$

where $\hat\mu$ denotes the unit vector in the $\mu$-direction and the sum is over the $D$ directions.

To define a lattice field theory action, we need to define a discrete version of all components in the continuous quantum field theory action. The integral is replaced by a sum

$$\int_{-\infty}^{\infty} d^D x \mapsto a^D \sum_{x \in V_L}, \tag{3}$$

and the derivatives by the difference

$$\partial_\mu \phi(x) \mapsto \frac{\phi_{x+\hat\mu a} - \phi_x}{a}. \tag{4}$$

Putting everything together, we obtain the desired discretized action

$$S(\phi) = a^D \sum_x \left( \frac{1}{2} \sum_{\mu=1}^d \frac{\left(\phi_{x+a\hat\mu} - \phi_x\right)^2}{a^2} + \frac{m^2}{2} \phi_x^2 + \frac{\lambda}{4!} \phi_x^4 \right), \tag{5}$$

where we use $\phi$ to denote the field configuration $\phi = \{\phi_x\}_{x \in V_L}$. The corresponding Boltzmann distribution is given as

$$p_{\text{phys}}(\phi) = \frac{e^{-S(\phi)}}{\mathcal{Z}},$$

where $\mathcal{Z} = \int \prod_x d\phi_x \, e^{-S(\phi)}$ is the constant that ensures $\int \prod_x d\phi_x \, p_{\text{phys}}(\phi) = 1$.

| | QFT | LFT |
|---|---|---|
| *spacetime* | $\int \mathrm{d}^D x$ | $a^D \sum_{x \in V_L}$ |
| *field* | $\phi : \mathbb{R}^D \to \mathbb{R}$ | $\phi : V_L \to \mathbb{R}$ |
| | $x \mapsto \phi(x)$ | $x \mapsto \phi_x$ |
| *derivative* | $\partial_\mu \phi(x)$ | $\frac{1}{a}\big(\phi_{x+a\hat{\mu}} - \phi_x\big)$ |
| *action* | $S[\phi(x)]$ | $S(\phi)$ |
| *gauge field* | $A_\mu(x)$ | $U_\mu(x)$ |
| *observable* | $\langle \mathcal{O} \rangle = \frac{1}{\mathcal{Z}} \int \mathcal{D}\phi\, e^{-S[\phi(x)]} \mathcal{O}(\phi(x))$ | $\langle \mathcal{O} \rangle = \mathbb{E}_{\phi \sim p_{\text{phys}}} \mathcal{O}(\phi)$ |
| | | $\sim \dfrac{1}{N_{\text{conf}}} \displaystyle\sum_{i=1}^{N_{\text{conf}}} \mathcal{O}\big(\phi^{(i)}\big),$ |

## 2.1  Computing Physical Quantities by Sampling

To compute observables in quantum field theory, one has to perform the following (infinite-dimensional) path integral

$$\langle \mathcal{O} \rangle_{\text{QFT}} = \frac{1}{\mathcal{Z}_{\text{cont.}}} \int \mathcal{D}\phi\, e^{-S_{\text{cont.}}[\phi]} \mathcal{O}[\phi] \equiv \lim_{\substack{a \to 0 \\ L \to \infty}} \frac{1}{\mathcal{Z}} \int \prod_x \mathrm{d}\phi_x\, e^{-S(\phi)} \mathcal{O}(\phi). \tag{6}$$

In lattice field theory, the observables are computed via the last expression of the above formula, without taking the continuum limit. To compute it, it is necessary to have enough samples drawn from the Boltzmann distribution $p_{\text{phys}}(\phi) = \frac{1}{\mathcal{Z}} e^{-S(\phi)}$. In the following subsection, we will discuss the traditional Monte Carlo-based sampling methods that can be employed to obtain samples.

Once equipped with the independently drawn samples $\{\phi^{(i)}\} = \{\phi_x^{(i)}\}_{x \in V_L}, i = 1, \ldots N_{\text{conf}}$, any observable can be evaluated, by computing its average

$$\langle \mathcal{O} \rangle = \mathbb{E}_{\phi \sim p_{\text{phys}}}[\mathcal{O}(\phi)] \equiv \lim_{N_{\text{conf}} \to \infty} \frac{1}{N_{\text{conf}}} \sum_{i=1}^{N_{\text{conf}}} \mathcal{O}(\phi^{(i)}), \tag{7}$$

where $N_{\text{conf}}$ is the size of the dataset.

## 2.2  Sampling

In this subsection we discuss the sampling methods, without the use of machine learning tools, that can be employed to sample from the Boltzmann distributions. The challenges of employing these methods will be discussed in the subsequent subsections.

### 2.2.1 Monte Carlo Methods

One way to generate configurations according to the Boltzmann distribution of the theory, $p_{\mathrm{phys}}(\phi) = \frac{1}{Z} e^{-S(\phi)}$, is to build a Markov chain by using an updating method to generate new configurations, often through making some local changes. Starting with a randomly chosen initial configuration, the updates are applied sequentially to move the system towards configurations that are consistent with the target distribution.

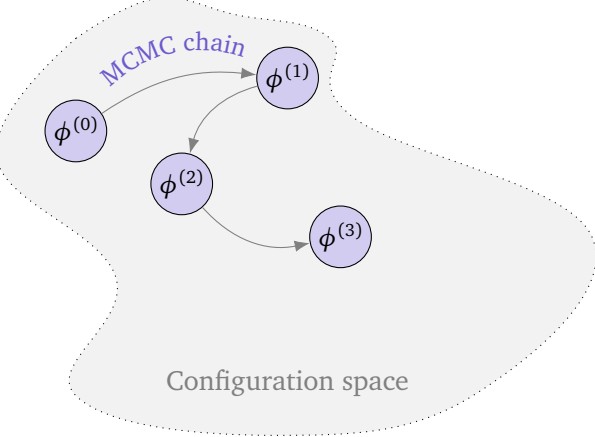

$$\phi_0 \to \phi_1 \to \cdots \to \phi_n \to \phi_{n+1} \to \ldots$$

During the process, the Markov chain explores the configuration space, guided by the updating scheme. Each configuration is drawn based on the current state and the chosen transition matrices $W_{\phi\phi'} := W(\phi \to \phi')$, satisfying the condition

$$\sum_{\phi'} W_{\phi\phi'} = 1$$

guaranteeing the conservation of probabilities. In order to have the guarantee that our process leads to samples approaching the target probability, the transition matrix $W$ needs to have certain additional properties:

**Ergodicity:** Any $\phi'$ can be reached from any other configuration $\phi$ within a finite number of Markov steps.

**Detailed balance:** At each step of the Markov process, the probability distribution $p^{(n)}(\phi)$ changes according to

$$\sum_{\phi} p^{(n)}(\phi) W_{\phi\phi'} = p^{(n+1)}(\phi').$$

We hence need to require the equilibrium condition

$$\sum_{\phi} p(\phi) W_{\phi\phi'} = p(\phi').$$

Namely, if the transition matrix is applied to an ensemble of samples of the target distribution, the newly generated ensemble will still be distributed according to the target distribution.

For the above to hold, the detailed balance condition is sufficient (although not necessary):

$$\frac{W_{\phi\phi'}}{W_{\phi'\phi}} = \frac{p(\phi')}{p(\phi)}.$$

After a certain number of steps, the initial random field configurations are increasingly mapped into those reflecting the target distribution. The time required for this is sometimes referred to as the thermalization period. After the thermalization phase, in which the generated samples are to be discarded, the system reaches equilibrium and the subsequent configurations generated by the Markov process are representatives of the desired distribution.

A popular Markov Chain Monte Carlo (MCMC) algorithm is *Metropolis-Hastings algorithm*, which involves accepting a new configuration according to a ratio of probabilities $r$, as shown in Algorithm 1[2].

---

[2]We consider the situation where the proposal samples are drawn independently.

**Algorithm 1:** Metropolis-Hastings algorithm

**Input:** $q(\phi)$: proposal density, $p(\phi)$: target density,
initial sample $\phi^{(0)}$ .
**Output:** $\phi^{(1)}, \phi^{(2)}, \ldots, \phi^{(N)}$

**for** $i = 1 : N$ **do**
  $\phi' \sim q(\phi)$
  $u \sim \text{Uniform}(0, 1)$
  $r = \min\left(1, \frac{q(\phi^{(i-1)})}{p(\phi^{(i-1)})} \frac{p(\phi')}{q(\phi')}\right)$
  **if** $u < r$ **then**
    $\phi^i = \phi'$                          ▷ Accept
  **else**
    $\phi^i = \phi^{i-1}$                     ▷ Reject

Since the *equilibrium condition* applies, given enough time, the history, or "trajectory", of the Markov Chain $\{\phi^{(0)}, \phi^{(1)}, \ldots, \phi^{(N)}\}$ will give a representative ensemble of the target distribution. Though equipped with this theoretical guarantee on the asymptotic exactness of the sampling process, the exploration of the configuration space using the Metropolis algorithm can be slow, and one might waste a lot of computational resources to reach equilibrium, especially when the dimension of the configuration space is high. The slow exploration results in large autocorrelations between the samples, and as a result the large errorbars for the estimators, as will be detailed in section 2.3.

### 2.2.2   Hybrid (or Hamiltonian) Monte Carlo Methods

Compared to Metropolis-Hastings algorithms that are based on random walks, the Hamiltonian (or hybrid) Monte Carlo (HMC) method is a specific type of Metropolis-Hastings sampling that can lead to a faster exploration of the configuration space. This is achieved through the use of a fictitious Hamiltonian evolution which is capable of proposing more distant configurations, while the approximate energy conservation (broken only by the numerical errors caused by the use of a discrete numerical integrator) guarantees that the proposal samples are very often accepted [3], resulting in a relatively efficient sampler that is the current go-to method for many lattice field theory computations.

In HMC, we augment the original $D$-dimensional space (the $\phi$-space) by adding the conjugate momenta, resulting in a $2D$-dimensional phase space (the $(\phi, \pi)$-space). Consider the Hamiltonian equations[3]

$$\frac{\mathrm{d}\pi}{\mathrm{d}t} = -\frac{\partial H(\phi, \pi)}{\partial \phi}, \quad \frac{\mathrm{d}\phi}{\mathrm{d}t} = \frac{\partial H(\phi, \pi)}{\partial \pi}, \tag{8}$$

for a Hamiltonian $H(\phi, \pi)$, and the corresponding Boltzmann distribution

$$P(\phi, \pi) = \frac{1}{\mathcal{Z}_H} \exp(-H(\phi, \pi)), \tag{9}$$

for $(\phi, \pi)$. The solutions of the Hamiltonian equations manifestly preserve the value of $H$, leading to movements confined to a hypersurface characterized by a constant probability density. Next, we choose the Hamiltonian to have the form $H(\phi, \pi) = V(\phi) + K(\pi)$, where $V$ is related to the target distribution via $p_{\text{target}}(\phi) = e^{-V(\phi)}/\mathcal{Z}$, so $V(\phi) = S(\phi)$ in the case of lattice field theories. By integrating out the momenta, the sample distribution approaches that of the target

---

[3]We put the mass parameter to 1.

distribution. Indeed, at its core, HMC constructs a combined distribution $P(\phi, \pi)$ and an evolution dynamics such that the trajectories lead to samples of the desired target distribution.

In the special case of the ordinary kinetic term $K(\pi) = \frac{1}{2}\pi^2$, HMC consists of the following steps:

1. sample an independent momentum variable $\pi$ distributed according to the Gaussian distribution $\exp\left(-\pi^2/2\right)/\mathcal{Z}_\pi$ and an initial $\phi$ ;

2. apply the dynamical evolution by numerically solving the Hamiltonian equations (8). This will generate some new coordinates in the phase space:

$$(\pi, \phi) \mapsto (\pi', \phi');$$

3. accept or reject the generated pair with probability

$$P_{\text{acc}} = \min\left[1, \exp\left(-\Delta H\right)\right], \quad \text{where} \quad \Delta H = H(\pi', \phi') - H(\pi, \phi), \tag{10}$$

as in the Metropolis algorithm.

A common choice for the numerical integration, is the *leapfrog integration*

$$(\phi, \pi) = (\phi(\tau), \pi(\tau)) \mapsto (\phi', \pi') = (\phi(\tau'), \pi'(\tau')).$$

In coordinates $(\phi^i, \pi_i)_{i=1,\dots,D}$, one step of the leapfrog integration is given by

$$\begin{aligned}
\pi_i\left(\tau + \frac{\epsilon}{2}\right) &= \pi_i(\tau) - \frac{\epsilon}{2} \left.\frac{\partial V(\phi)}{\partial \phi^i}\right|_{\phi=\phi(t)} \\
\phi_i(\tau + \epsilon) &= \phi_i(\tau) + \epsilon \pi_i\left(\tau + \frac{\epsilon}{2}\right) \\
\pi_i(\tau + \epsilon) &= \pi_i\left(\tau + \frac{\epsilon}{2}\right) - \frac{\epsilon}{2} \left.\frac{\partial V(\phi)}{\partial \phi^i}\right|_{\phi=\phi(t+\epsilon)}.
\end{aligned} \tag{11}$$

Due to the discrete nature of the above numerical integration, there can be small mismatch between the energy of the start and final configurations. These errors can accumulate over the trajectory, leading to a larger mismatch. Nevertheless, the correct asymptotic distribution is guaranteed by the additional Metropolis-Hastings acceptance-reject step, which is also responsible for the stochastic element of the HMC method together with the initial sampling step of the momentum. By tuning the step size $\epsilon$, one can balance between high acceptance (smaller $\epsilon$) and faster exploration (larger $\epsilon$), and exert control on the correlations between samples in the HMC algorithm.

## 2.3 Critical Slowing Down

A key challenge in LFT is to efficiently generate field configurations $\{\phi^{(i)}\}$, with the main obstacle being the phenomenon of critical slowing down [4]. This tends to happen when the system approaches the continuum limit or a second order phase transition. In the former case, the physical size $L \times a$ is kept fixed while the lattice spacing is sent to zero $a \to 0$. In the latter case, the correlation length diverges, $\xi/a \to \infty$. In both of these limits, the number of the effective degrees of freedom becomes very large. Subsequently, samples tend to become significantly correlated, resulting in a slow exploration of the large configuration space.

Given an observable, we now define quantities that measure how correlated samples are in a Markov chain.

**Autocorrelations**   Denoting by $A_t$ the measurement of the observable $A$ evaluated at the $t$-th configuration $\phi_t$ in a given Markov chain, the autocorrelation between measurements is shown to exhibit an exponential decay as the sample time interval increases [5]:

$$\langle A_t A_{t+\Delta t}\rangle - \langle A\rangle^2 \propto \exp\left(-\Delta t/\tau_A^{(\mathrm{exp})}\right). \tag{12}$$

A commonly used quantity is the *integrated autocorrelation time*:

$$\tau_A := \frac{1}{2} + \frac{\sum_{\Delta t=1}^{\infty}\left(\langle A_t A_{t+\Delta t}\rangle - \langle A\rangle^2\right)}{\langle A^2\rangle - \langle A\rangle^2}. \tag{13}$$

To see its significance, consider $N$ successive observations $A_t$, the average of the variance is given by

$$(\Delta A)^2 = \left\langle (\bar{A} - \langle A\rangle)^2\right\rangle = \left\langle \left(\frac{1}{N}\sum_{t=1}^{N} A_t - \langle A\rangle\right)^2\right\rangle \tag{14}$$

$$= \frac{1}{N^2}\sum_{t=1}^{N}\left(\langle A_t^2\rangle - \langle A\rangle^2\right) + \frac{2}{N^2}\sum_{t=1}^{N}\sum_{\Delta=1}^{N-t}\left(\langle A_t A_{t+\Delta}\rangle - \langle A\rangle^2\right) \tag{15}$$

$$\approx \frac{1}{N}\operatorname{Var}A \cdot 2\tau_A, \tag{16}$$

where $\operatorname{Var}A = \langle A^2\rangle - \langle A\rangle^2$ is the variance of the observable. In comparison, for $N$ *uncorrelated* samples, the $1\sigma$ error bar for large enough $N$ is

$$\Delta A = \sqrt{\frac{\operatorname{Var}A}{N}}. \tag{17}$$

From this we see that the quantity $2\tau_A$ represents the number of update sweeps required to make the samples statistically independent from the starting configuration.

As argued heuristically above, the quantity $2\tau_A$ can become large close to a phase transition. This phenomenon of *critical slowing down* is captured by the relation

$$\tau_A \propto \left[\min(\xi/a, L)\right]^z, \tag{18}$$

taking into account the finiteness of the lattice, where $z$ a dynamical critical exponent that is typically about 2 for local update methods. This means that the number of iterations necessary to generate a new configuration grows as the correlation length ($\xi$) or the lattice size ($L$) increases. In these regimes, it is important to develop algorithms that can mitigate critical slowing down, by incorporating appropriate global moves (as opposed to only relying on local moves such as individual spin-flips) for example. For the Ising model, the Wolff and Swendsen-Wang algorithm have been particularly successful, but it is desirable to develop techniques that will be applicable to a broader range of theories. Finally, if the theory has non-trivial topological properties, such as gauge theories, the system is subject to another, more severe type of critical slowing down called the topological freezing. This will be discussed in section 2.4 and section 2.5.

## 2.4   Topological Freezing

Earlier we have seen that simulating lattice field theories can be a challenging task, particularly near the continuum limit or near phase transitions, as illustrated by the critical slowing down

phenomenon (18), where one typically has $z \simeq 2$ [6, 4, 7, 8]. Intuitively, this is a consequence of the random walk behavior where the distance between the original location of particle and the location after $\tau$ steps is proportional to $\sqrt{\tau}$. As a result, for a simulation to move through a distance given by the correlation length $\xi$, it requires $\mathcal{O}((\xi/a)^2)$ steps. In the presence of non-trivial topological structure in a quantum field theory, things become even more complicated. The autocorrelation of a topological observables $A$, for which the random walk intuition is not applicable, does not need to satisfy the power law specified in (18). In a gauge theory, for instance, the above scaling (18) does not hold for topological observables such as a topological charge, for which the scaling is observed to be exponential [7, 9]. For instance, one observes [7, 10]

$$\tau(Q^2) \sim \exp\left(c(\xi/a)^\theta\right),\tag{19}$$

where $\theta$ has a value reported to be in the range $0.4 - 0.5$ [7, 11, 10] is a universal critical exponent and $Q$ the lattice topological charge that we will define in (147) later, though it must also be said that it is challenging to numerically differentiate between exponential behavior and power law behavior with a very large exponent. At any rate, regardless of the uncertainties regarding the exact functional form of the critical slowing down of the topological charge, it is evident that the critical slowing down phenomenon is significantly more severe for topological observables. This imposes stringent constraints on the lattice size that can be realistically simulated.

The effect of the severe slowing down described above can be understood in terms of topological sectors. There is, strictly speaking, no disjoint topological sectors on a finite lattice as the energy barrier remains finite. That said, the definition of topological charges on a lattice allows for the segmentation of configuration space into distinct sectors, with the energy barriers between them growing with $\xi/a$ [12]. The elevated barriers suppress the tunneling rate within a Markov Chain dictated by a local updating algorithm, causing the system to become trapped in a single fixed topological sector. This phenomenon, known as *topological freezing*, disrupts ergodicity and leads to significant systematic errors. It limits the precise calculation of topological observables such as the topological charge or the associated susceptibility, as reflected in the scaling of the autocorrelation time described above.

In practice, this constitutes a serious challenge in lattice QCD, where interesting physical phenomena are linked to topological properties, including those associated to chiral symmetry breaking, the strong CP problem [13, 14], and the large mass of the $\eta'$ meson [15]. As we will review later, normalizing flows offer a promising framework for overcoming the topological critical slowing down, due to its nature as an independent sampler which samples from all topological sectors indiscriminately.

## 2.5 An Example: $\mathbb{C}P^{N-1}$ Model

The $\mathbb{C}P^{N-1}$ model [16] is helpful in understanding certain aspects of quantum chromodynamics, including asymptotic freedom, confinement and a non-trivial vacuum structure with stable instanton solutions [17] . It serves as an interesting toy model for the Yang-Mills theory, since it is fermion-free and nevertheless exhibits topological freezing, providing testing ground for computational algorithms.

A 2-dimensional $\mathbb{C}P^{N-1}$ model is defined by the action [18]

$$S[z(x)] = \frac{1}{g} \int \mathrm{d}^2 x \, \overline{D_\mu z}(x) \cdot D_\mu z(x),\tag{20}$$

where $z : \mathbb{R}^2 \to \mathbb{C}^N$ is a $N$-component complex scalar field, constrained to lie on the sphere

$S^{2N-1}$

$$\bar{z}(x)z(x) = \sum_{i=1}^{N} \bar{z}^i(x)z^i(x) = 1 \tag{21}$$

and $D_\mu$ is the covariant derivative given by

$$D_\mu = \partial_\mu + iA_\mu. \tag{22}$$

The gauge equivalence of the theory is given by $(z, A) \sim (z', A')$ when the pairs satisfy

$$z^{i'}(x) = e^{i\Lambda(x)}z^i(x), A'_\mu(x) = A_\mu(x) - \partial_\mu \Lambda(x), \tag{23}$$

for some local $U(1)$ transformations specified by $e^{i\Lambda} : \mathbb{R}^2 \to U(1)$.

The manifold $\mathbb{C}P^{N-1}$

$$S^{2N-1}/U(1) \simeq \mathbb{C}P^{N-1} \tag{24}$$

can be defined as the space of all complex lines in $\mathbb{C}^N$ passing through the origin, with real dimension $2(N-1)$.

Since the action contains no kinetic term for $A_\mu$, it can be eliminated using the following equation of motion

$$A_\mu(x) = \frac{i}{2}\left(\bar{z}\partial_\mu z - z\partial_\mu \bar{z}\right). \tag{25}$$

The model also possesses a global $SU(N)$ symmetry

$$z \mapsto Uz, \quad U \in SU(N). \tag{26}$$

In what follows we explore some aspects of the theory that are of interest to us.

**Instantons** The existence of locally stable non-trivial minima of the action stems from the inequality

$$\int d^2x \left| D_\mu z(x) \mp i \sum_\nu \epsilon_{\mu\nu} D_\nu z(x) \right|^2 \geq 0, \tag{27}$$

where $\epsilon_{12} = -\epsilon_{21} = 1$ and $\epsilon_{\mu\mu} = 0$, the equality holds if and only if the (anti)self-duality equation

$$D_\mu z(x) = \pm i \sum_\nu \epsilon_{\mu\nu} D_\nu z(x) \tag{28}$$

is satisfied. Defining

$$Q[z(x)] = -i \sum_{\mu,\nu} \epsilon_{\mu\nu} \int d^2x \, D_\mu z(x)\overline{D_\nu z}(x) = i \sum_{\mu,\nu} \int d^2x \, \epsilon_{\mu\nu}(D_\nu D_\mu z(x))\bar{z}(x) \tag{29}$$

and expanding the above expression, one finds

$$|Q| \leq g\,S. \tag{30}$$

Using

$$i\sum_{\mu,\nu} \epsilon_{\mu\nu} D_\mu D_\nu = \frac{1}{2}i\sum_{\mu,\nu} \epsilon_{\mu\nu}[D_\nu, D_\mu] = -\frac{1}{2}\sum_{\mu,\nu} \epsilon_{\mu\nu} F_{\mu\nu}, \tag{31}$$

where $F_{\mu\nu}(x) = -i[D_\nu, D_\mu] = \partial_\mu A_\nu(x) - \partial_\nu A_\mu(x)$ is the $U(1)$ field strength, the topological charge can be expressed as

$$Q = -\frac{1}{2} \sum_{\mu,\nu} \int d^2x \, \epsilon_{\mu\nu} F_{\mu\nu}(x) = -\lim_{R \to \infty} \oint_{|x|=R} d\boldsymbol{x} \, \boldsymbol{A}(x). \tag{32}$$

The above expression makes manifest that the charge only depends on the behavior at asymptotic infinity ($|x| \to \infty$). Finiteness of the action demands that at large radius $D_\mu z(x)$, and hence $F_{\mu\nu}$, vanishes, and $A_\mu$ is a pure gauge given by (cf. (23))

$$A_\mu(x) = \partial_\mu \Lambda(x), \, z(x) = e^{i\Lambda(x)} z_* \text{ at } |x| \to \infty,$$

for some $z_* \in \mathbb{C}^N$ and $\Lambda(x)$. From the above we see that the topological charge $Q = 2\pi n$ measures the winding number $n \in \mathbb{Z}$ of the map

$$x \in S^1 \mapsto e^{i\Lambda(x)} \in U(1) \simeq S^1, \tag{33}$$

which can be thought of as an element of the first homotopy group $\pi_1(S^1) = \mathbb{Z}$. From the discussion before, we have $S \geq 2\pi|n|/g$ where the equality corresponds to a local minimum which holds only for field configurations satisfying the (anti)self-duality equation (28). The corresponding classical solution is the so-called instanton solutions of the $\mathbb{C}P^{N-1}$ model.

Writing the topological charge (32) as $Q = \int d^2x \, q(x)$, we can define the *susceptibility* $\chi_t$ as

$$\chi_t = \int d^2x \, \langle q(x)q(0) \rangle = \frac{\langle Q^2 \rangle}{V}. \tag{34}$$

It measures the amount of topological excitations of the vacuum and is a renormalization group invariant quantity.

**Lattice** The lattice equivalent of the covariant derivative is [19, 20]

$$D_\mu z(x) := \frac{1}{a} \left( U_\mu(x) z(x + a\hat{\mu}) - z(x) \right), \quad \text{with} \quad \overline{U}_\mu(x) U_\mu(x) = \mathbb{1}. \tag{35}$$

In the continuum limit, $U_\mu(x)$ can be expressed via the gauge field as

$$U_\mu(x) = e^{iaA_\mu(x)}. \tag{36}$$

In what follows, we will put $a = 1$ to simplify expressions. The action (20) is recovered from the lattice action

$$S = \frac{1}{g} \sum_{n,\mu} \overline{D_\mu z_n} \cdot D_\mu z_n. \tag{37}$$

Gauge transformations on a lattice, under which the action is invariant, is given by

$$U_\mu(x) \to e^{i\Lambda(x)} U_\mu(x) e^{-i\Lambda(x+\hat{\mu})}, \quad z^j(x) \to e^{i\Lambda(x)} z^j(x), \tag{38}$$

for some $e^{i\Lambda(x)} : V_L \to U(1)$, where $V_L$ again denotes the set of lattice points.

From the lattice version of the action

$$S = -\frac{1}{g} \sum_{x,\mu} (\overline{U}_\mu(x)\bar{z}(x+\hat{\mu})z(x) + U_\mu(x)\bar{z}(x)z(x+\hat{\mu})),$$

we see that the partition function is given by

$$\mathcal{Z} = \int DU \int Dz \exp\left( \frac{1}{g} \sum_{x,\mu} (\overline{U}_\mu(x)\bar{z}(x+\hat{\mu})z(x) + U_\mu(x)\bar{z}(x)z(x+\hat{\mu})) \right), \tag{39}$$

where a field independent term has been discarded, and

$$DU \equiv \prod_{\mu,x} dU_\mu(x), \qquad Dz \equiv \prod_{x,j} \frac{d\bar{z}(x)^j \, dz(x)^j}{2\pi i} \delta(\bar{z}z - 1), \tag{40}$$

with $dU_\mu(x)$ being the $U(1)$ Haar measure.

As usual, all non-vanishing physical observables must be invariant under the gauge transformations (see §4.1.3). Consequently, we can consider the local gauge-invariant composite operator, since the fields $z, \bar{z}$ themselves are not invariant. Let

$$P(x) = \bar{z}(x) \otimes z(x), \tag{41}$$

or

$$P_{ij}(x) = \bar{z}_i(x)z_j(x) \tag{42}$$

in the matrix notation. Its group invariant correlation function[4] is given by

$$G_P(x) = \mathrm{Tr}\langle P(x)P(0)\rangle_{\mathrm{conn}} = \mathrm{Tr}\langle P(x)P(0)\rangle - \frac{1}{N}. \tag{43}$$

From this we can define the two point susceptibility

$$\chi_2 = \sum_{x\in V_L} G_P(x). \tag{44}$$

Summing over one of the two dimensions, we define

$$G_s(x_2) = \sum_{x_1=1}^{L} G_P(x_1, x_2). \tag{45}$$

From the expectation that $G_s(x_2) \sim e^{-x_2/\xi}$, we note

$$\frac{G_s(x_2+1) + G_s(x_2-1)}{2G_s(x_2)} \sim \frac{e^{-\frac{1}{\xi}} + e^{\frac{1}{\xi}}}{2} = \cosh\frac{1}{\xi}.$$

Consequently, we can define the inverse correlation length as

$$\frac{1}{\xi} = \frac{1}{L-1} \sum_{x_2=1}^{L-1} \mathrm{arcosh}\left( \frac{G_s(x_2+1) + G_s(x_2-1)}{2G_s(x_2)} \right). \tag{46}$$

To define the lattice counterpart of the topological charge density (32), note that a naive translation of $\mathrm{Tr}(F^\star F)$ on the lattice often leads to non-integer-valued topological charge due to short-distance effects. For instance, in the continuum $Q$ can be expressed as the integral of a total derivative and does not renormalize, properties that do not hold on a lattice [21].

---

[4]We have used $\langle P_{ij}(x)\rangle = \frac{1}{N}\delta_{ij}$, which implies $\mathrm{Tr}\left(\langle P_{ij}(x)\rangle\langle P_{ij}(0)\rangle\right) = \frac{1}{N}$.

The following, the so-called geometrical, definition of the lattice counterpart of topological charge (32), has the virtue that it gives rise to an integer-valued topological charge $Q$. It is given by $Q_L = \sum_{x \in V_L} q_L(x)$ [22], with

$$q_L(x) \equiv \frac{1}{4\pi} \epsilon_{\mu\nu} \big( \theta_\mu(x) + \theta_\nu(x + \hat{\mu}) - \theta_\mu(x + \hat{\nu}) - \theta_\nu(x) \big) \mod 1; \quad -\frac{1}{2} < q_L(x) \le \frac{1}{2}, \quad (47)$$

and $\theta_\mu(x) = \arg(\bar{z}(x)z(x + \hat{\mu})) \in (-\pi, \pi]$.

Another definition of the topological charge density is (see for instance [23])

$$q_L(x) = -\frac{i}{2\pi} \sum_{\mu\nu} \epsilon_{\mu\nu} \text{Tr} \big[ P(x) \Delta_\mu P(x) \Delta_\nu P(x) \big], \quad (48)$$

with

$$\Delta_\mu P(x) := \frac{P(x + \hat{\mu}) - P(x - \hat{\mu})}{2}. \quad (49)$$

The product $P(x)\Delta_\mu P(x)\Delta_\nu P(x)$ combines the field configuration and its differences in both directions $\mu$ and $\nu$ to capture how the field 'twists' in the lattice. Similarly, eq. (47) detects the winding of the phase around a plaquette and is also a local, gauge invariant quantity.

Over the years, there have been many proposals on how to simulate the $\mathbb{C}P^{N-1}$ model. Despite employing a sophisticated over-heat bath algorithm in [24, 25, 26], it was discovered that the simulations of topological observables experience exponential critical slowing down. Cluster algorithms have also been proposed in [27], but unlike the Ising and $O(N)$ model, they do not seem to mitigate the problem of critical slowing down and in fact show no improvement compared to a local Metropolis algorithm. In [28], it was argued that the reason for the lack of improvement lies in the fact that the cluster reflection in this case has codimension larger than one, unlike the case of the $O(N)$ models.

This limitation poses an obstacle for conducting further non-perturbative calculations to compare with the large $N$ and continuum predictions. Finally, the worm algorithm has been applied in [29], which does not suffer from topological slowing down, but the introduction of the new 'flux variables' makes the reconstruction of topological quantities difficult.

It is suggested [30, 31, 32] that with open (Neumann) boundary conditions in the physical time direction, the topological sectors disappear and the space of smooth fields becomes connected. Then, topological freezing is avoided, but autocorrelation times are still large. Allowing the inflow/outflow of topology with open boundary conditions, however, introduces boundary effects, limits the available space-time volume, and breaks translational invariance. To circumvent the latter problem it has been proposed [33] to use a non-orientable manifold instead.

Given the above, the $\mathbb{C}P^{N-1}$ model, while easier to simulate due to its low-dimensions, presents significant challenges due to the afore-mentioned topological freezing, making it an excellent testing ground for new simulation methodologies such as gauge-equivariant flow-based algorithms.

# 3   Normalizing Flows

Generative AI has experienced revolutionary progress in recent years, leading to myriad applications with transformative potential. In this lecture note we focus on the normalizing flow approach. In this section we introduce different normalizing-flow architectures, keeping in mind their applications to lattice quantum field theories.

$$z = z(0) \sim r(z) \qquad\qquad \phi = z(T) := f(z(0)) \sim q_f(\phi)$$

**Figure 1:** A map from the latent space, where the density distribution takes a simple form, to the configuration space where the physical theory lives. The map provides a diffeomorphism between the two spaces.

## 3.1   Trivializing a Physical Theory

Recently, novel sampling methods based on normalizing flows have emerged as a promising approach to tackle the issue of critical slowing down in lattice field theories [34, 35, 36, 37, 38, 39, 40, 41, 35, 42, 43, 44, 45]. The main idea, in physical terms, is to construct a map that "trivializes" the theory, converting the target theory into a 'simpler' theory where the degrees of freedom are disentangled. The non-trivial aspects of the theory are then encoded in the invertible map. In the case of flow-based machine learning models, the task of finding, or rather approximating, a trivializing map is formulated as an optimization problem.

In principle, this method has the potential to surpass traditional sampling techniques in efficiency, as sample proposals are drawn independently. In practice, the story is more complicated as one also needs to take into account the difficulties and the computational costs of learning and using such a trivializing map.

Concretely, let's say our goal is to generate samples $\left\{\phi^{(1)}, \phi^{(2)}, \ldots, \phi^{(N)}\right\}$ of the random field variables $\phi_x$ on the lattice $V_L$, i.e. $\phi = \{\phi_x : x \in V_L\}$, such that they are distributed according to the Boltzmann distribution

$$p(\phi) = \frac{e^{-S(\phi)}}{\mathcal{Z}}, \quad \text{where} \quad \mathcal{Z} = \int \prod_{x \in V_L} d\phi_x \, e^{-S(\phi)}. \tag{50}$$

In the context of flow-based models, the objective is to find a bijective $f$, such that it has the effect of mapping the probability a distribution of interest, $p(\phi)$, into an easy-to-sample distribution, $r(z)$, often corresponding to a trivial or free theory. This is illustrated in fig. 1, for the special case where the map $f$ is chosen to be given by integrating an ODE.

In the machine learning setup, this map $f = f_\theta : z \mapsto \phi$ is parametrized by a set trainable parameters, collectively denoted by $\theta$, that are optimized such as the pushforward density (cf. (51)) approximates the target distribution $p(\phi)$.

$$q_f(\phi) \approx p(\phi).$$

The quality of the approximation depends on a variety of factors ranging from the neural network architecture to the training procedure, as usual in machine learning. But even if the model is not perfectly trained, $q_f(\phi)$ can be helpful as a good proposal distribution in conjunction with the Metropolis-Hastings algorithm (see Algorithm 1) for a traditional MCMC sampling method, in importance sampling, and more. In this way, the success of the flow-based sampling method relies on how easy it is to compute the pushforward density (cf. (51)) accurately, and how well it approximates the target density. We will discuss these points in more detail in the remaining of the section and in §6.

## 3.2   Introducing Normalizing Flows

Normalizing flows [46, 47, 48, 49] aim to provide an invertible deterministic map that transforms a simple distribution (such as a normal distribution, the namesake of normalizing flows) into a complex, multi-modal "target" distribution. Equipped with such a map, one can draw samples from the simple distribution, and their images under the map constitute samples drawn from the target distribution. Different normalizing flow architectures provide different neural-network parametrizations of the invertible map.

A normalizing flow must satisfy a crucial property: the transformation $f$ must be not only invertible but also differentiable. This means the latent space and the configuration space are diffeomorphic, and in particular have the same dimension.

Under these conditions, given a latent space density (the prior), the density on the configuration space is given by

$$q_f(\phi) = r(z)\big|\det J_f(z)\big|^{-1}. \tag{51}$$

From the above, we see that in the physical case, requiring the transformed density coincides with physical probability density $p(\phi) = e^{-S(\phi)}/\mathcal{Z}$ is to require the map $f$ to satisfy

$$S(f(z)) - \log|\det J_f(z)| + \log r(z) = \text{constant}. \tag{52}$$

---

**Explanation:** Given a diffeomorphism $f$, the local change of volume given by the change of variables is given by $\mathrm{d}^d f = df_1(z) \wedge \cdots \wedge df_d(z) = \big|\det J_f(z)\big|\mathrm{d}^d z$, in terms of the Jacobian matrix

$$J_f(z) = \frac{\partial f(z)}{\partial z} = \begin{pmatrix} \frac{\partial f_1}{\partial z_1} & \cdots & \frac{\partial f_1}{\partial z_d} \\ \vdots & \ddots & \vdots \\ \frac{\partial f_d}{\partial z_1} & \cdots & \frac{\partial f_d}{\partial z_d} \end{pmatrix}.$$

From this we see that (51) satisfies $q(\phi)\,\mathrm{d}\phi = r(z)\,\mathrm{d}z$.

---

Now, given that the composition of two diffeomorphisms leads to a diffeomorphism, with

$$(f_2 \circ f_1)^{-1} = f_1^{-1} \circ f_2^{-1},$$
$$\det J_{f_2 \circ f_1}(z) = \det J_{f_2}(f_1(z)) \cdot \det J_{f_1}(z)$$

we can apply a series of transformations $f_{i \in \{1,\dots,k\}}$ to generate the normalizing flow (see fig. 2)

$$z_0 \sim r(z), \tag{53}$$
$$z_k = f(z_0) = f_k \circ \cdots \circ f_1(z_0), \tag{54}$$

with $z = z_0$ and $z_k = \phi$. Applying (51) at every step, we can flow from the original distribution $q_0(z_0) = r(z)$ to the target distribution $q_k(z_k) = p(\phi)$,

$$z_k \sim q_k(z_k) = q_0(z_0) \prod_{i=1}^{k} \left| \frac{\partial f_i(z_{i-1})}{\partial z_{i-1}} \right|^{-1}, \quad \text{with } z_i = f_i \circ \cdots \circ f_1(z_0). \tag{55}$$

The possibility to concatenate maps to form more complicated maps is a key concept in the design of this type of generative models, and leads to the "flow" part of the name "normalizing flows".

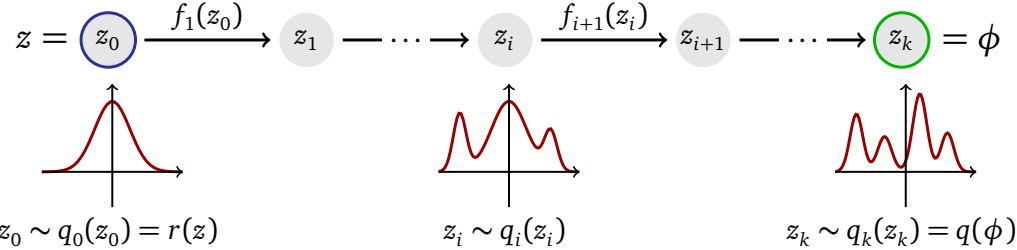

**Figure 2:** A normalizing flow maps the latent variable $z_0 \sim r(z)$ to the configuration space variable $\phi \sim q(\phi)$

As mentioned before, a flow-based model performs two main functions: sampling from the model according to

$$\phi = f(z), \quad \text{where} \quad z \sim r(z), \tag{56}$$

and computing the model's probability density using eq. (51), facilitating it to approximate the target density. These lead to different computational considerations. For the former, it is necessary that one is able to easily sample from the initial distribution $r(z)$ and compute the forward transformation $f$. For the latter, evaluating the model's density requires computing the determinant of the Jacobian. Different implementations of normalizing flows provide different tradeoffs in the above-mentioned considerations, some of which will be mentioned in the remainder of the section. Additional techniques developed to improve normalizing flows include stochastic normalizing flows, flow-matching, and more.

### 3.2.1   Training the Normalizing Flows

As often the case in machine learning, training a flow-based model is an optimization process, aiming to have the model distribution $q_{f_\theta}(\phi)$ matching a target distribution $p(\phi)$, by minimizing a measure of the difference between the two, using optimization techniques such as stochastic gradient descent.

**KL divergence**

A common choice for the loss function is given by the Kullback-Leibler (KL) divergence

$$\begin{aligned}
\text{KL}(p \,\|\, q) &= \int \mathcal{D}\phi \; p(\phi) \log \frac{p(\phi)}{q(\phi)} \\
&= \mathbb{E}_{\phi \sim p}[\log p(\phi)] - \mathbb{E}_{\phi \sim p}[\log q(\phi)] = -H(p) + H(p, q),
\end{aligned} \tag{57}$$

which is a combination of an entropy and a cross entropy term and provides a measure of the difference between two probability distributions. Note that it does not serve as a metric in the space of probability functionals, since it is neither symmetric $\text{KL}(q \,\|\, p) \neq \text{KL}(p \,\|\, q)$ nor satisfies the triangular inequality. It does have the property of positive semi-definiteness, and vanishes if and only if $p = q$, making it a reasonable choice of loss function.

**Proof:** We will prove the non-negative property of the KL divergence, which is equivalent to Gibb's inequality, using that $\log x \leq x - 1$ for $x > 0$ and the equality holds only when

$x = 1$. From this it follows

$$
\begin{aligned}
-\mathrm{KL}(p \parallel q) &= -\int \mathcal{D}\phi\, p(\phi) \log \frac{p(\phi)}{q(\phi)} \\
&= \int \mathcal{D}\phi\, p(\phi) \log \frac{q(\phi)}{p(\phi)} \\
&\leq \int \mathcal{D}\phi\, p(\phi) \left( \frac{q(\phi)}{p(\phi)} - 1 \right) \\
&= \int \mathcal{D}\phi\, q(\phi) - \int \mathcal{D}\phi\, p(\phi) \\
&= 1 - 1 = 0
\end{aligned}
$$

and $\mathrm{KL}(p \parallel q) = 0$ only if $p = q$.

**Reverse KL** The reverse KL divergence between a proposal distribution $q$ and the target distribution $p$ is given by exchanging $p$ and $q$ in (57)

$$
\begin{aligned}
\mathrm{KL}(q \parallel p) &= \int \mathcal{D}\phi\; q(\phi) \log \frac{q(\phi)}{p(\phi)} \\
&= \mathbb{E}_q[\log q(\phi)] - \mathbb{E}_q[\log p(\phi)] = -H(q) + H(q,p).
\end{aligned}
\tag{58}
$$

In the above, $H(q)$ denotes the entropy and $H(q,p)$ the cross-entropy. Note that, while a priori a useful measure of the difference between $p$ and $q$, care must be taken when using it as a loss function. This is because its minimization tends to promote the "mode-seeking" behaviour, encouraging $q$ to vanish in regions where $p$ vanishes but not necessarily encouraging $q$ to be non-vanishing when $p$ does not vanish, as the latter type of discrepancy tends to be insufficiently punished by an increase of the loss function since the expectation is taken with respect to $q$.

Note that, to compute the reverse KL divergence, it is not necessary to have samples drawn from the target distribution. On the other hand, one needs to be able to calculate the probability of the true model $p(\phi)$ up to its normalization factor. Such a factor can be ignored as it only contributes an additive constant and is irrelevant for the optimization. This property makes reverse KL is a convenient choice of loss function for lattice field theory applications. In such applications, the target distribution $p$ takes the form $p(\phi) = \frac{1}{\mathcal{Z}} e^{-S(\phi)}$. When the proposal distribution is modeled through a normalizing flow map $f$ by (51), we get

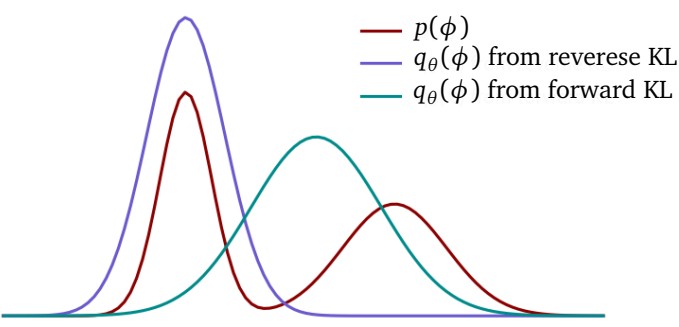

**Figure 3:** An illustration of a typical optimization results using forward and reverse KL.

$$
\mathrm{KL}(q \parallel p) = \log \mathcal{Z} - H(r) + \mathbb{E}_{z \sim r} L(z), \quad \text{with } L(z) = S(f(z)) - \log|\det J_f(z)|. \tag{59}
$$

Clearly, the last term is the only term that is relevant for optimization.

**Forward KL** Similarly, we have

$$\mathrm{KL}(p \parallel q) = -H(p) + H(p, q). \tag{60}$$

and minimizing the forward KL is equivalent to minimizing the cross entropy $H(p, q)$, the negative log likelihood $-\mathbb{E}_{\phi \sim p}[\log q(\phi)]$.

Hence, optimizing using forward KL tends to lead to $q$ that has non-vanishing mass wherever $p$ has mass. This leads to the so-called "mean-seeking" behavior, where all modes of the target distribution are to some extent covered by the proposal distribution. Unlike the reverse KL, the computation of forward KL divergence requires a dataset of samples drawn from the true model $p(x)$.

## 3.3 Coupling Layers

Another family of bijective transformations is the coupling flows [50, 51]. They have the advantage of having a triangular Jacobian and therefore a tractable Jacobian determinant. A common building block for these models is the affine coupling layers.

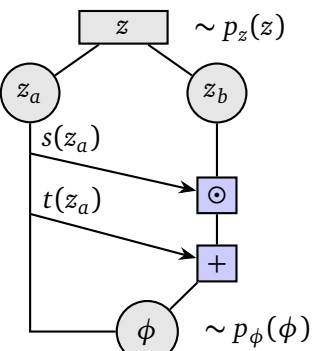

In each such layer, where we have the map $f : z \in \mathbb{R}^N \mapsto \phi \in \mathbb{R}^N$, the degrees of freedom of the input variable $z = (z_1, z_2, \ldots, z_N)$ is split into two equal-sized subsets $N_a$, $N_b$ (the passive and active parts respectively) which are transformed according to

$$
\begin{aligned}
\phi_a &= z_a, \\
\phi_b &= z_b s_b(z_a) + t_b(z_a)
\end{aligned}
\quad \Leftrightarrow \quad
\begin{aligned}
z_a &= \phi_a, \\
z_b &= (\phi_b - t_b(\phi_a))/s_b(\phi_a),
\end{aligned}
$$

where $s_b, t_b$ are neural networks.

**Figure 4:** A coupling layer.

The Jacobian is given by the lower-triangle matrix

$$J_f = \frac{\partial \phi}{\partial z} = \begin{pmatrix} \mathbb{I}_a & 0 \\ \frac{\partial \phi_b}{\partial z_a} & \mathrm{diag}(s_b(z_a)) \end{pmatrix}, \tag{61}$$

and therefore has tractable determinant.

Note that the transformation of a coupling layer leaves the components in $N_a$ unchanged. A more general transformation can be obtained by composing coupling layers in an alternating pattern, ensuring that the components left unchanged in one coupling layer are updated in the next. Since the determinant is multiplicative ($\det AB = \det A \det B$), each coupling layer $\ell$, chosen to have an alternating assignment of active and passive components, simply adds to a summand to

$$\log|\det J_f(z)| = \sum_\ell \sum_{b_\ell} \log|s_{b_\ell}^{(\ell)}(z_{a_\ell})|. \tag{62}$$

One drawback of the coupling layer is that the partition of the sample space into two prevents equivariance under the full symmetry group of the lattice, a property that we will discuss in the next subsection. See [34, 52] for the applications of coupling layers for simulating the lattice $\phi^4$ theory. Their applications to gauge theories will be discussed in §6.2.1.

## 3.4 Continuous Flows

### 3.4.1 Residual Flows

Residual flows [53] are maps built from residual layers, that are functions of the form

$$\mathbf{y} \equiv F(\mathbf{x}) = \mathbf{x} + g(\mathbf{x}; \theta). \tag{63}$$

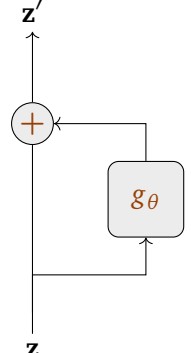

$\mathbf{z}'$

$\mathbf{z}$

**Figure 5:** A residual flow building block.

One motivation for such a design is the fact that these maps are manifestly free from the problems of vanishing gradients, as its Jacobian matrix is given by

$$\frac{\partial \mathbf{y}}{\partial \mathbf{x}} = \mathbf{I} + J_g,$$

where $J_g$ is the Jacobian of the map $g$, or

$$\frac{\partial y^i}{\partial x^j} = \delta^i_j + \frac{\partial g^i(\mathbf{x}; \theta)}{\partial x^j}$$

in the component form. Moreover, it is easy to choose the function $g$ such that the map $F$ is invertible.

The change of the density is then given by (51)

$$\log p_{\mathbf{x}}(\mathbf{x}) = \log p_{\mathbf{y}}(\mathbf{y}) + \log \left| \det\left(\mathbf{I} + J_g(x)\right) \right| = \log p_{\mathbf{y}}(\mathbf{y}) + \operatorname{tr} \log\left(\mathbf{I} + J_g(x)\right), \tag{64}$$

which can be approximated with Taylor expansion

$$\operatorname{tr} \log (\mathbf{I} + J) = \sum_{n=1}^{\infty} \frac{(-1)^{n+1}}{n} \operatorname{tr}(J^n), \tag{65}$$

if $\|J\|_2 < 1$.

To gain expressivity, one would like to build a deep residual network by stacking up the residual layers mentioned above. However, this means that we need to compute the Jacobian determinant (64) for each residual layer. The computation using the Taylor expansion by truncating terms beyond a given power is an approximation and leads to error that can accumulate. Furthermore, one also needs to compute the backpropagation in order to train. In principle, this is a straightforward task, but the memory cost it incurs can be high.

### 3.4.2 Neural ODEs

An approach that avoids the afore-mentioned difficulties of deep residual networks is the neural ODEs [54]. One way to view it is as the infinite-depth limit of deep residual networks, but parametrized by a finite number of learnable parameters. We will see how, exploiting the associated ODEs, the backpropagation and the change of densities can be computed relatively easily, for instance by ODE integrators with an adaptive number of integration steps.

To see this explicitly, note that the transformation

$$\mathbf{z}_{t+1} = \mathbf{z_t} + v(\mathbf{z}(t), t; \theta)\Delta t, \tag{66}$$

becomes an ODE

$$\frac{d\mathbf{z(t)}}{dt} = v(\mathbf{z}(t), t; \theta), \tag{67}$$

when the limit of $\Delta t \to 0$ is taken. One would like to optimize the parameters $\theta$ parametrizing the normalizing map $f$ by minimizing (the average over $\mathbf{z}(0) \sim r(z)$) a loss function

$$L\big(\mathbf{z}(t_f)\big) = L\left(\mathbf{z}(t_i) + \int_{t_i}^{t_f} dt\, v(\mathbf{z}(t), t; \theta)\right). \tag{68}$$

A powerful method, called the **adjoint sensitivity method**, provides a way to backpropagate through the ODE. One way to describe it is by first introducing a Lagrange functional for the ODE (67)

$$\mathcal{J} = L\big(\mathbf{z}(t_f)\big) + \int_{t_i}^{t_f} dt\, a(t)\,(\dot{\mathbf{z}}(t) - v(\mathbf{z}(t), t; \theta)), \tag{69}$$

with a Lagrange multiplier $a(t)$ enforcing the ODE. Then one can show the following proposition

**Proposition 1** Assuming that $a(t)$ satisfies the ODE

$$\frac{d a(t)}{dt} = -a(t)^T \frac{\partial v(\mathbf{z}(t), t, \theta)}{\partial \mathbf{z}(t)} \tag{70}$$

and the boundary condition

$$a(t_f) = -\frac{\partial L}{\partial \mathbf{z}(t_f)}, \tag{71}$$

then one has

$$\frac{d\mathcal{J}}{d\theta} = -\int_{t_i}^{t_f} dt\, a(t) \frac{\partial v(\mathbf{z}(t), t; \theta)}{\partial \theta}. \tag{72}$$

*Proof:* Here we give an informal proof. In terms of components, denoting $\mathbf{z} = (z^1, z^2, \ldots, z^d)$ and similarly for $a$. From the chain rule we have

$$\frac{d\mathcal{J}}{d\theta} = \frac{\partial L}{\partial z^i(t_f)} \frac{\partial z^i(t_f)}{\partial \theta} + \int_{t_i}^{t_f} dt\, a_i(t)\left(\frac{\partial \dot{z}^i}{\partial \theta} - \frac{\partial v^i}{\partial \theta} - \frac{\partial z^j}{\partial \theta}\frac{\partial v^i}{\partial z^j}\right)(t)$$

$$= \frac{\partial L}{\partial z^i(t_f)} \frac{\partial z^i(t_f)}{\partial \theta} + a_i \frac{\partial z^i}{\partial \theta}\bigg|_{t_i}^{t_f} - \int_{t_i}^{t_f} dt\, \frac{\partial z^j(t_f)}{\partial \theta}\left(\dot{a}_j + a_i \frac{\partial v^i}{\partial z^j}\right)(t) \tag{73}$$

$$- \int_{t_i}^{t_f} dt\, a_i(t)\frac{\partial v^i}{\partial \theta},$$

where we have integrated by part going from the first to the second and the third line, and a sum over the repeated indices is implied as usual. Noting that the first two terms of the second line cancel since $\frac{\partial z^i(t_i)}{\partial \theta} = 0$ as a part of the setup of the problem, and due to the boundary condition at $t_f$ (71). The last term in the second line also vanishes when the adjoint field $a(t)$ satisfies the ODE (70), and only the expression in the last line remains.

A more formal derivation using the language of differential geometry can be found in the Appendix A of [43].

From the proposition, we see that the gradient of the loss function (68) in terms of the solution

to the ODE (67) can be obtained by performing the following integration

$$
\boldsymbol{a}(t) = \boldsymbol{a}(t_f) + \int_t^{t_f} \mathrm{d}t' \, \boldsymbol{a}(t')^T \, \frac{\partial \boldsymbol{v}(\mathbf{z}(t'), t', \theta)}{\partial \mathbf{z}(t')},
$$
$$
\frac{\mathrm{d}L}{\mathrm{d}\theta} = -\int_{t_i}^{t_f} \mathrm{d}t \, \boldsymbol{a}(t) \frac{\partial \boldsymbol{v}(\mathbf{z}(t), t; \theta)}{\partial \theta},
\tag{74}
$$

which can be done by one single call of the ODE solver.

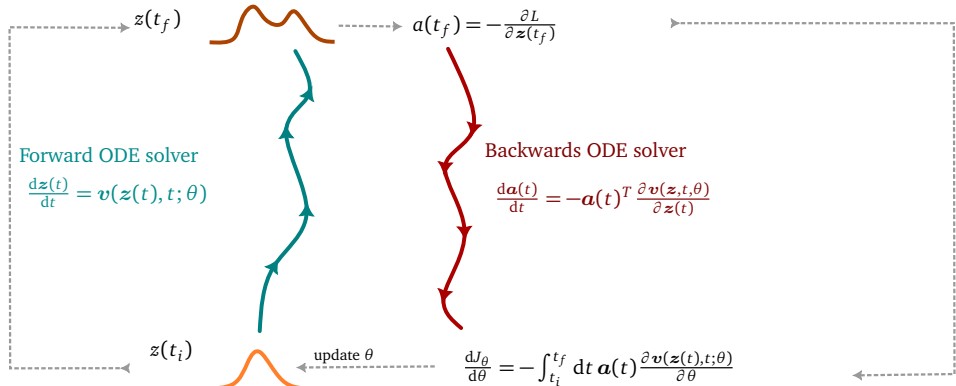

**Figure 6:** Neural ODE algorithm.

Finally, another important feature of neural ODE is that the change of density can also be obtained by solving an ODE. Consider an infinitesimal time interval $\mathrm{d}t$, one has

$$
\mathbf{z}_{t+\mathrm{d}t} = \mathbf{z}_t + \boldsymbol{v}(\mathbf{z}_t)\,\mathrm{d}t + \mathcal{O}\!\left(\mathrm{d}t^2\right)
\tag{75}
$$

and hence (cf. (64))

$$
\log p_{t+\mathrm{d}t}(z_{t+\mathrm{d}t}) = \log p_t(z_t) - \mathrm{d}t \, \mathrm{tr}(J_{\boldsymbol{v}}) + \mathcal{O}\!\left(\mathrm{d}t^2\right),
\tag{76}
$$

leading to

$$
\frac{\mathrm{d}}{\mathrm{d}t} \log p_t(z(t)) = -\mathrm{tr}(J_{\boldsymbol{v}}) = -\nabla \cdot \boldsymbol{v}(z(t), t; \theta).
\tag{77}
$$

in the limit $\mathrm{d}t \to 0$.

Equation (77) describes one of the main advantages of Neural ODEs. The often costly computation of the determinant of the Jacobian factor in (51) is replaced by the computation of a trace. Another advantage is their flexibility. The neural network $\boldsymbol{v}(\mathbf{z}(t), t; \theta))$ can be parametrized in almost any desired way; since uniqueness of the path is guaranteed, the transformation will be automatically bijective, as long as the appropriate Lipschitz condition is met. Thanks to its flexibility, it is often easy to incorporate the desired equavariance properties of the network in neural ODEs. The parametrization of the vector field $\boldsymbol{v}$ whose trace is easy to compute is particularly desirable due to the role of the latter in the density ODE (77). On the other hand, the ODE integrations can be computationally expensive, which might result into a longer training and sampling time compared to other neural network architectures.

Neural ODE flows have been employed in [39, 40] for simulating the lattice $\phi^4$ theory. Their applications to gauge theories will be discussed in §6.2.3.

## 3.5 Equivariance

In generative models and in other machine learning contexts, it is often beneficial to preserve the inherent structure of the datasets (the so-called "inductive bias"), especially the symmetries. This is achieved through the so-called equivariance property, ensuring that symmetries in the input space are meaningfully and consistently reflected in the output space, of the neural networks. Explicitly, if the input and output spaces of a map $f$ both furnish representations, denoted $\rho_{\text{in}}$ and $\rho_{\text{out}}$ respectively, of a symmetry group $G$, then the map $f$ is said to be *equivariant* if

$$f(\rho_{\text{in}}(g)x) = \rho_{\text{out}}(g)f(x) \quad \forall \ g \in G, \tag{78}$$

for all elements $x$ of the input space. This definition is depicted in Figure 7. Equivariant neural networks have been extensively explored in the literature. See [55, 56, 57, 58, 59, 60] for some examples.

Alternatively, one can have a model that does not have any built-in equivariance and hope that the model learns the symmetries during training. When data efficiency is of importance, the general expectation is however that incorporating the symmetries from the outset can result in more data-efficient training, and often more robust generations. For example, a model that is equivariant under rotational symmetry can more easily generate desired results for a rotated version of the inputs even if it has seen fewer samples in the rotated orientation during training.

Finally, physical theories are rich in symmetries, and the equivariance of the network is therefore an important consideration in physics applications. In the context of normalizing flows, if the prior distribution obeys the symmetries under consideration, then under the equivariant diffeomorphic mapping satisfying (78) with in this case $\rho_{\text{in}} = \rho_{\text{out}} = \rho$, then the push-forward (model) distribution given by (51) will also have the same symmetry

$$r(z) = r(\rho(g)z) \Rightarrow q_f(\rho(g)f(z)) = q_f(f(z)). \tag{79}$$

The flexibility of neural ODE makes it easy to build in symmetries in this architecture; one simply needs to make sure that the vector field $v(z(t), t; \theta)$ satisfies the equivariant condition (78) for arbitrary parameters $\theta$.

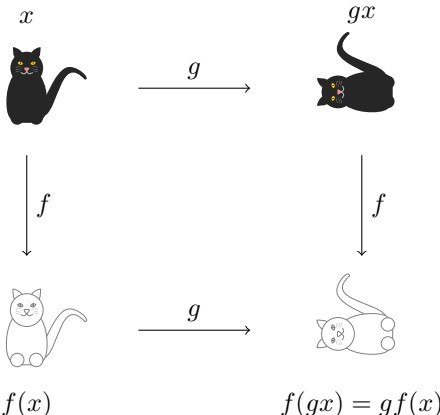

**Figure 7:** Equivariance in the form of a commutative diagram (78).

In what follows we discuss some examples of symmetries relevant for physical theories, in the context of continuous normalizing flows.

**Example: Lattice Symmetries**

The symmetries of a two-dimensional square lattice with periodic boundary condition in both directions ($V_L \cong (\mathbb{Z}/L\mathbb{Z})^2$) is

$$G_{\text{latt}} := C_L^2 \rtimes D_4 \,, \tag{80}$$

the semidirect product of two cyclic groups $C_L^2$ (translations) with the dihedral group $D_4$ (rotations and reflections). Such spatial symmetries can easily be incorporated using the so-called convolutional kernels [61, 62, 63, 64], satisfying

$$w : V_L \times V_L \to \mathbb{R} \ \text{ with } \ w_{x,y} = w_{gx,gy} \ \forall g \in G_{\text{latt}}. \tag{81}$$

We shall see shortly how the above can be incorporated in neural ODEs.

**Example: $\mathbb{Z}_2$ Global Symmetries in Scalar Theories**

In [39, 40], following [65], equivariance has been employed to construct a shallow network architecture. Consider the scalar $\phi^4$ theory (5), which has the global $\mathbb{Z}/2$ symmetry acting as $\phi_x \mapsto -\phi_x$, leaving the action invariant.

Using the convolutional kernel discussed above and introducing the time feature $K(t)_d$ to encode explicit time dependence of the vector field, we can parametrize the neural ODE as

$$\frac{\mathrm{d}\phi_x(t)}{\mathrm{d}t} = \sum_{y,d,f} W_{x,y,d,f} K(t)_d H_f(\phi_y(t)), \tag{82}$$

where $d = 1,\ldots D$, $f = 1,\ldots F$, and $W \in \mathbb{R}^{L^2 \times L^2 \times D \times F}$ the learnable weight tensor with a convolutional structure (81).

In order to be equivariant with respect to the $\mathbb{Z}/2$ symmetry, the $\phi$-feature must be odd under it: $H(-\phi) = -H(\phi)$. In [39, 40], we chose $H_f(\phi) = \sin(\omega_f \phi)$ or $H_f(\phi) = \phi$ with learnable $\omega_f$. As a result, the neural ODE (82) is equivariant under the full symmetries, including the global symmetry and the lattice symmetries, of the lattice $\phi^4$ theory.

**Example: Gauge Equivariance**

Finally, we would like to consider gauge symmetries, where the group transformation depends on the spacetime location. Rather than a symmetry of the theory, a convenient point of view is to consider these as redundancies in our description of the theory, which should be taken into account carefully and consistently. In the next section we will discuss the physics of lattice gauge theories and in §6 the flow-based sampling methods for these theories.

## 4 Lattice Gauge Theories

An important feature of the Standard Model is the role of the quark fields. In this section we introduce the basics of lattice field theories with gauge fields.

### 4.1 Yang-Mills Theory on a Lattice

In this section we introduce the fundamental degrees of freedom, the gauge link variables, of lattice gauge theories. First we give a lightning overview of gauge theories in continuous spacetime, before presenting the lattice counterpart and discussing the effects of discretization.

### 4.1.1   Gauge Fields

To appreciate the role of the gauge fields, consider vectors of fermionic fields

$$\psi(x) = \begin{pmatrix} \psi_1(x) \\ \psi_2(x) \\ \vdots \\ \psi_N(x) \end{pmatrix} \in \mathbb{C}^N. \tag{83}$$

with the free action

$$S_F^{\text{free}} = \int \mathrm{d}^d x \, \bar{\psi}(x)(\gamma^\mu \partial_\mu + m)\psi(x). \tag{84}$$

This theory has global $SU(N)$ symmetries, transforming the quark fields as

$$\psi(x) \mapsto \Omega\psi(x) \quad \text{and} \quad \bar{\psi}(x) \mapsto \bar{\psi}(x)\Omega^\dagger, \quad \text{where} \quad \Omega \in SU(N). \tag{85}$$

The above global symmetries can be promoted to local symmetries with the introduction of gauge fields $A_\mu(x) \in \mathfrak{su}(N)$, and including them in the action

$$S_F = \int \mathrm{d}^d x \, \bar{\psi}(x)(\gamma^\mu D_\mu + m)\psi(x), \tag{86}$$

with $D_\mu := \partial_\mu + iA_\mu$. Indeed, observe that the above action possesses now a local "gauge" symmetry

$$\psi(x) \mapsto \Omega(x)\psi(x), \ \bar{\psi}(x) \mapsto \bar{\psi}(x)\Omega^\dagger(x), \ A_\mu(x) \mapsto \Omega(x)A_\mu(x)\Omega^\dagger(x) + i(\partial_\mu\Omega)\Omega^\dagger(x), \tag{87}$$

where $\Omega \in SU(N)$. We will often express the gauge field $A_\mu = \sum_a T_a A_\mu^a$ in the Lie algebra basis $\{T_a\}$.

The natural kinetic term is the Yang-Mills action

$$S_{\text{YM}} = -\frac{1}{2g^2} \int d^d x \, \text{Tr}\big(F_{\mu\nu}F^{\mu\nu}\big), \tag{88}$$

expressed in terms of its field strength $F_{\mu\nu} := i[D_\mu, D_\nu]$. Note that $F_{\mu\nu}(x)$ transforms as

$$F_{\mu\nu}(x) \mapsto \text{Conj}_{\Omega(x)}(F_{\mu\nu}(x)),$$

where we have denoted the conjugation transformation, $\text{Conj} : G \to \text{Aut}(G)$, as

$$\text{Conj}_X(U) := XUX^{-1}, \ \text{for} \ X, U \in G, \tag{89}$$

and the Yang-Mills action is as a result invariant under gauge transformations. As mentioned above, as opposed to global symmetries, gauge symmetries can be understood as the redundancies of the description of the physics in terms of the degrees of freedom we use to write down the local Lagrangian: all field configurations related by a gauge transformation of the form (87) should be regarded as physically equivalent, and redundant degrees of freedom can be removed via "gauge fixing" using (87).

The gauge field $A_\mu$ can be studied from a geometric point of view. Just as the Christoffel symbol $\Gamma_{\nu\rho}^\mu$ in general relativity dictates how a test particle with a small mass should be parallel transported from an initial point $x_i$ in spacetime to a final point $x_f$, the gauge field dictates how a

test particle with a "color charge" should be parallel transported from one point to the other. And just as the Christoffel symbol, gauge fields can be described in the geometric language of fiber bundles.

To be more explicit, to parallel transport between fibers at different points along a path $\mathcal{C}$ connecting $x_i$ and $x_f$, one uses the connection[5]

$$\Phi(x_f) = U(x_i, x_f; \mathcal{C})\Phi(x_i), \quad \text{where} \quad U(x_i, x_f; \mathcal{C}) = \mathcal{P} \exp\left( i \int_\mathcal{C} dx^\mu A_\mu \right), \qquad (90)$$

where $\mathcal{P}$ denotes path ordering. The above operator $U(x_i, x_f; \mathcal{C})$ is called the *Wilson line* operator and transforms as

$$U(x_i, x_f; \mathcal{C}) \mapsto \Omega(x_i)U(x_i, x_f; \mathcal{C})\Omega^\dagger(x_f) \qquad (91)$$

under gauge transformation (87). Note that $U(x_i, x_f; \mathcal{C})$ depends on not just the endpoints but also the path $\mathcal{C}$, and it is not necessarily identity when $\mathcal{C}$ is a closed loop. Instead, $U(x_i, x_f = x_i; \mathcal{C})$ captures the holonomy of the connection around the closed loop $\mathcal{C}$ and is called the *Wilson loop* operator. It transforms as

$$U(x, x; \mathcal{C}) := \mathcal{P} \exp\left( i \oint_\mathcal{C} dx^\mu A_\mu \right) \mapsto \text{Conj}_{\Omega(x)}(U(x, x; \mathcal{C})), \qquad (92)$$

under the gauge transformation (87), and its trace is therefore gauge invariant.

Now consider putting the theory on a square lattice of dimension $d$, $|V_L| = L^d$, and associate a value of a quark field $\psi(x)$ to each lattice point $x \in V_L$. In order to parallel transport between different lattice points, from the above discussion we see that we need to introduce a Wilson line operator for each directed edge of the lattice. Like in (2), we denote the starting and end points of the edges by $x$ and $x \pm a\hat{\mu}$, with $\hat{\mu}$ denotes the different directions of the lattice and $a$ the lattice spacing: we have

$$U_\mu(x) := U(x, x + a\hat{\mu}; \mathcal{C}) = \mathcal{P} \exp\left( i \int_x^{x+a\hat{\mu}} A_\mu(z) dz^\mu \right), \qquad (93)$$

where $\mathcal{C}$ is the directed edge pointing from the lattice point $x$ to the lattice point $x + a\hat{\mu}$. As a result, in the continuum limit, the the Wilson line operator is given as

$$\lim_{a \to 0} U_\mu(x) = e^{iaA_\mu(x)}(1 + O(a)). \qquad (94)$$

From (93), we also have

$$U_{-\mu}(x) := U_\mu^\dagger(x - \hat{\mu})$$

as illustrated in the following figure.

After introducing these fundamental degrees of freedom, we would like to write down the lattice counterpart of the Yang-Mills action (88). To capture $F_{\mu\nu} = -i[D_\mu, D_\nu]$, let us introduce the plaquette operator, which is a special case of the Wilson loop operator, of the form

$$P_{\mu,\nu}(x) = U_\mu(x)U_\nu(x + \hat{\mu})U_\mu^\dagger(x + \hat{\nu})U_\nu^\dagger(x). \qquad (95)$$

[5]In this lecture notes we only consider matter fields transforming in fundamental representation, although an analogous statement holds for more general representations.

We can rewrite the plaquette $P_{\mu,\nu}(x)$ in terms of $A_\mu$ using the Campbell-Hausdorff formula: $\exp(A)\exp(B) = \exp\left(A + B + \frac{1}{2}[A,B] + \dots\right)$ and the Taylor expansion

$$A_\nu(x + \hat{\mu}) = A_\nu(x) + a\partial_\mu A_\nu(x) + \mathcal{O}(a^2), \tag{96}$$

which gives

$$P_{\mu,\nu}(x) = e^{ia^2 F_{\mu\nu}(x) + O(a^3)}. \tag{97}$$

A simple gauge-invariant action in terms of the plaquette Wilson loops is thus

$$S(U) = \frac{\beta}{N}\sum_{x \in V_L}\sum_{\mu < \nu}\mathrm{Re}\,\mathrm{Tr}\left[\mathbb{1} - P_{\mu,\nu}(x)\right] = \frac{\beta}{2N}\sum_{x \in V_L}\sum_{\mu,\nu}\mathrm{Tr}\left[\mathbb{1} - P_{\mu,\nu}(x)\right], \tag{98}$$

where the sum in the second expression includes plaquettes of all orientations, and we used $\mathrm{Re}\,\mathrm{Tr}\left(P_{\mu,\nu}\right) = \frac{1}{2}\mathrm{Tr}\left(P_{\mu,\nu} + P_{\mu,\nu}^\dagger\right)$ in the last equality.

Using the relation (97) with the continuous Yang-Mills variables, we see

$$\mathrm{Tr}\left[\mathbb{1} - P_{\mu,\nu}(x)\right] = \mathrm{Tr}\left(\mathbb{1} - e^{ia^2 F_{\mu\nu}(x) + \dots}\right) = \mathrm{Tr}\left(ia^2 F_{\mu\nu} - \frac{a^4}{2}(F_{\mu\nu})^2 + \dots\right) = -\frac{a^4}{2}\mathrm{Tr}\left(F_{\mu\nu}(x)^2\right) + \dots$$

and hence

$$S(U) = \frac{\beta a^4}{4N}\sum_{x \in V_L}\sum_{\mu,\nu}\mathrm{Tr}\left(F_{\mu\nu}(x)^2\right) + \mathcal{O}\left(a^6\right). \tag{99}$$

A comparison with the continuous Yang-Mills action gives the relation between the bare couplings

$$\beta = \frac{2N}{g^2}.$$

Another action which is often used is the Wilson action [66]

$$S(U) = -\frac{\beta}{2N}\sum_{x \in V_L}\sum_{\mu,\nu}\mathrm{Tr}\left[P_{\mu,\nu}(x)\right], \tag{100}$$

which only differs from (98) by an irrelevant constant.

### 4.1.2   Improved Gauge Action

Inevitably, any discretization introduces errors, and the Wilson action (100) deviates from Yang-Mills action at higher ($O(a^6)$) order. One way to address it is the so-called Symanzik improvements [67, 68, 69], where one includes higher-order terms in the action to eliminate the discretization errors, order by order. A different approach is the idea of renormalization group, where higher energy degrees of freedom are integrated over and absorbed in the dynamics of the discretized, "blocked" fields. In this approach, it would be ideal to design lattice actions that completely eliminate artifacts at all orders. These are known as quantum perfect actions, and they are in general challenging to find. A more modest and realistic approach is to exploit

the idea of Wilsonian RG and design the so-called *fixed-point* actions [70] that are free from lattice artifacts at the classical level. The classical predictions on the lattice, using a classical fixed point action, should agree with those in the continuum. Finding such a fixed-point action is believed to help in achieving more accurate results in the continuous limit even when simulated on a coarser lattice. See for instance [71] for a machine learning approach in finding such a fixed-point action.

Here we discuss the improved action from the first viewpoint. As the Wilson action only coincides with the Yang-Mills action at order $a^4$. Controlling the $O(a^6)$ effects in simulations by adding extra terms to the Wilson gauge action can help reduce discretization errors without needing excessively fine lattice spacings.

In [72], it was argued that an improvement at the next leading order can be achieved by appropriately incorporating three types of Wilson loops with six edges that are denoted by $\mathcal{W}_1$, $\mathcal{W}_2$, and $\mathcal{W}_3$ in Figure 8, apart from the plaquette Wilson loops $\mathcal{W}_0$. The resulting action is sometimes called the Lüscher-Weisz action.

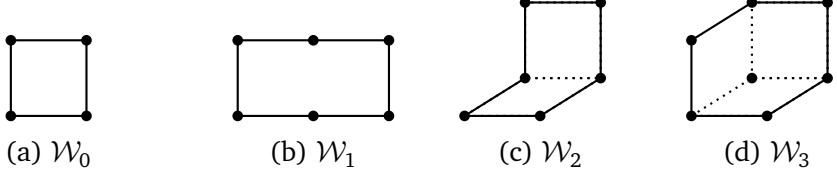

(a) $\mathcal{W}_0$        (b) $\mathcal{W}_1$        (c) $\mathcal{W}_2$        (d) $\mathcal{W}_3$

**Figure 8:** Types of Wilson loops with four and six edges.

Relatedly, one can consider the improvement of the discrete counterpart of the field strength $F_{\mu\nu}$. A popular choice of discretization is

$$\hat{F}_{\mu\nu}(x) = -\frac{i}{8a^2}\left(Q_{\mu\nu}(x) - Q_{\mu\nu}^\dagger - \frac{1}{N}\operatorname{Tr}\left(Q_{\mu\nu} - Q_{\mu\nu}^\dagger\right)\right) + \mathcal{O}(a^2), \tag{101}$$

with $Q_{\mu\nu}$ given by the following sum of plaquette Wilson loop operators (95) starting and ending at $x$:

$$Q_{\mu\nu} = P_{\mu,\nu}(x) + P_{\nu,-\mu}(x) + P_{-\mu,-\nu}(x) + P_{-\nu,\mu}(x). \tag{102}$$

Explicitly, besides (95) one has also

$$P_{\nu,-\mu}(x) := U_\nu(x,t)U_\mu^\dagger(x-\hat{\mu}+\hat{\nu},t)U_\nu^\dagger(x-\hat{\mu},t)U_\mu(x-\hat{\mu},t)$$

and similarly for the rest. Due to the shape of the Wilson loops involved in $Q_{\mu,\nu}$, the improvements terms in the fermionic action using the above $\hat{F}_{\mu\nu}$ are referred to as the clover improvements.

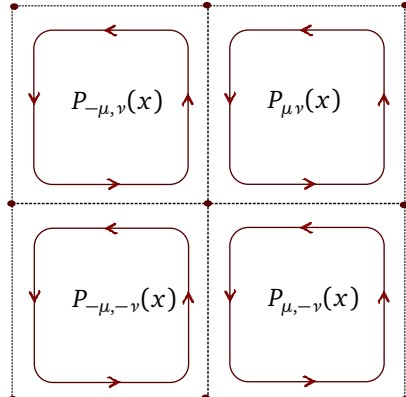

**Figure 9:** Clover-shaped discretization of the lattice field strength tensor

To show that $\hat{F}_{\mu\nu} \to F_{\mu\nu}$ in the continuum limit $a \to 0$, one performs the same calculation leading to (97) to obtain

$$
\begin{aligned}
Q_{\mu\nu}(x) &= 4 \cdot \exp\left(ia^2 F_{\mu\nu} + \mathcal{O}(a^4)\right) \\
&\simeq 4(1 + ia^2 F_{\mu\nu} + \mathcal{O}(a^4)).
\end{aligned}
\tag{103}
$$

In particular, the averaging over the four directions eliminates the discrepancy with the continuous limit at the order $\mathcal{O}(a^3)$. The definition (101) then makes sure that $\hat{F}_{\mu\nu}$ has the same anti-symmetry as $F_{\mu\nu}$, is traceless and hence lies in the Lie algebra $\mathfrak{su}(N)$, and has the right limit when $a \to 0$.

**Smearing**   Another effect of discretization is the numerical instability due to the artifact that the degrees of freedom are "pixelated". An additional improvement can hence be achieved with ultraviolet filtering, or smearing, of the gauge links $U_\mu$ in the action. This "fattening" of the gauge links mitigates cutoff effects, while preserving long-distance behaviour. for instance, it has been observed to have the effect of reducing the chiral symmetry breaking of Wilson quarks among light flavors [73, 74], and reducing flavor symmetry-breaking errors inherent in staggered fermions [75, 76].

Furthermore, as discussed in §2.5 and §4.2.5, there is no unique definition of topological charge density on the lattice. Different methods can be used to define and compute this quantity, but these methods might yield different results for the same lattice configuration. The discrepancy arises because the lattice, being a discrete structure, lacks the smoothness of the continuum. To address the ambiguities and obtain a meaningful topological charge density, filtering techniques are often employed. These techniques aim to "smooth out" the lattice configurations to better approximate the continuum fields and reduce the impact of short-range fluctuations, which are artifacts of the lattice discretization rather than true physical features. Smearing is one such technique.

Some common smearing methods include the stout smearing [77], the HYP smearing [78] and the HEX smearing [79]. Interestingly, the transformations which will be discussed in §6.2.3 can be viewed as (machine-learned) generalizations of the stout smearing.

**APE smearing**   Given a link, we consider the staples around it, which are combinations of neighboring links with shortest lengths (i.e. with three edges) and the same starting and end points and which obtains the name from its shape. APE smearing [80] seeks to replace each link with a weighted average between the original link and the sum of the staples around it.

Let $C_\mu(x)$ denote the following sum of staples:

$$C_\mu(x) = \sum_{\nu \neq \mu} \omega_{\mu\nu} \left( U_\nu(x) U_\mu(x+\hat{\nu}) U_\nu^\dagger(x+\hat{\mu}) + U_\nu^\dagger(x-\hat{\nu}) U_\mu(x-\hat{\nu}) U_\nu(x-\hat{\nu}+\hat{\mu}) \right), \quad (104)$$

where $\omega_{\mu\nu}$ are real, tunable parameters. For simplicity, we have put the lattice spacing to be $a = 1$ in the above equation and from now on, as it can be easily restored by dimensional analysis. Then,

$$V_\mu(x) = \mathcal{N}\left[ (1-\alpha) U_\mu(x) + \frac{\alpha}{2(D-1)} C_\mu(x) \right], \quad (105)$$

where $\alpha$ is a tunable parameter that controls the amount of smearing, and $\mathcal{N}$ is the normalization operator given by $\mathcal{N}[X] = \frac{X}{\sqrt{X^\dagger X}}$.

Equation (105) can be depicted as, in four dimensions,

$$\longrightarrow = \mathcal{N}\left[ (1-\alpha) \longrightarrow + \frac{\alpha}{6} \sum_\nu \left( \ulcorner\urcorner + \llcorner\lrcorner \right) \right].$$

However, the resulting smeared link $V_\mu(x)$ may not be an element of the original gauge group and a projection back onto the gauge group is typically required. The projection ensures that the smeared link remains a proper $SU(N)$ matrix, given by

$$U_\mu'(x) := \text{Proj}_{SU(N)}[V_\mu(x)]. \quad (106)$$

One way to perform this projection is by finding the closest group element $U_\mu'(x)$ to the smeared link $V_\mu(x)$ according to some matrix norm. In practice, this is usually done by maximizing the trace of the following product

$$U_\mu'(x) = \arg\max_{X \in G} \text{Re} \, \text{Tr}\left[ X V_\mu^\dagger(x) \right]. \quad (107)$$

The projection operation renders the APE procedure not differentiable and non-analytic, making it difficult to incorporate it in algorithms where the derivative with respect to the original gauge link $U_\mu(x)$ is required.

**Stout smearing**  A method of smearing link variables, which is analytic and hence differentiable, is stout smearing [77].

Using an isotropic smearing tunable parameter $\omega$, stout-link smearing involves a simultaneous update of all links on the lattice. Each link is replaced by a smeared link $U_\mu'(x)$, with

$$U_\mu'(x) = \exp\left( i X_\mu(x) \right) U_\mu(x), \quad (108)$$

where

$$X_\mu(x) = \frac{i}{2} \left( Q_\mu(x)^\dagger - Q_\mu(x) \right) - \frac{i}{2N} \text{Tr}\left( Q_\mu(x)^\dagger - Q_\mu(x) \right) \mathbf{1}. \quad (109)$$

Note that the trace term ensures that $X_\mu(x)$ belongs to the Lie algebra, and thus $\exp\left( i X_\mu(x) \right)$ to the Lie Group, eliminating the need for a projection operator. Furthermore,

$$Q_\mu(x) = \omega \sum \left\{ \text{plaquettes involving } U_\mu^\dagger(x) \right\} = C_\mu(x) U_\mu^\dagger(x), \quad (110)$$

where $C_\mu$ is the staple sum as in (104) as depicted in Figure 10 and no summation over $\mu$.

Specifically, the stout smearing parameter satisfy $\omega_{ij} = \omega$ and $\omega_{4\mu} = \omega_{\mu 4} = 0$ for the isotropic three-dimensional smearing, or $\omega_{\mu\nu} = \omega$ in the four-dimensional isotropic case. As can be

easily checked and as we will discuss in more details in §6, the above stout transformation is compatible with the gauge transformation (91).

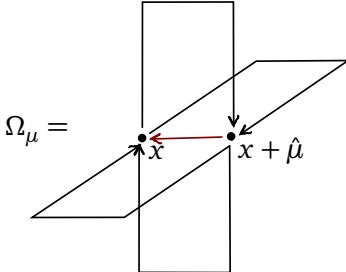

$\Omega_\mu =$

**Figure 10:** $\Omega_\mu = C_\mu(x)U_\mu^\dagger(x)$

The smeared link $U_\mu'(x)$ (108) is often called 'fat' link variable and can be schematically expressed as:

$$\longrightarrow = \left(e^{\,\square}\right) \longrightarrow$$
$$= \longrightarrow + \left(e^{\,\square} - 1\right) \longrightarrow \tag{111}$$

In §6.2.3 we will introduce a "trivializing map" in the context of simulating gauge fields. In fact, as noted by Lüscher in [12] and further examined in [81], trivializing maps can be thought of as being constructed from infinitesimal stout link smearing steps. Moreover, various gauge equivariant flow transformations of the gauge links we will discuss in §6 can be viewed as a machine-learning generalization of the stout smearing.

### 4.1.3   The Path Integral

As mentioned earlier, in lattice gauge theories we have the link variable $U_\mu(x)$, which takes value in the gauge group, instead of the gauge field $A_\mu(x)$ which takes value in the Lie algebra, as the fundamental degrees of freedom. In order to define the lattice partition function, we hence need to define the integration measure

$$\prod_{x,\mu} dU_\mu(x). \tag{112}$$

We need a measure for the group manifold which is invariant under right or left group multiplications:

$$\int dU\, f(\Omega U) = \int dU\, f(U\Omega') = \int dU\, f(U). \tag{113}$$

For compact, simple gauge groups, the *Haar measure* is defined to be a measure with the above symmetry property. It is moreover unique up to an overall multiplicative factor, which we fix by requiring

$$\int dU = 1. \tag{114}$$

As mentioned before, the integration $\prod dU_\mu(x)$ overcounts physical configurations; it integrates uniformly over all possible link configurations – including those that are related by a gauge transformation. While we usually deal with gauge redundancies with gauge fixing schemes in a continuous quantum field theory, the redundancy is not that problematic in a LFT because we

are dealing with finite dimensional integrals, and gauge fields $U_\mu(x)$ as well as the gauge transformations $\Omega(x)$ lie in compact spaces[6]. A common approach is therefore to simply integrate over all configurations on a lattice, including those that are gauge equivalent.

Then, the partition function of a pure gauge theory is given by

$$\mathcal{Z} = \int DU \, e^{-S(U)}, \quad DU := \prod_{x \in V_L} \prod_{\mu=1}^{d} \mathrm{d}U_\mu(x). \tag{115}$$

For example, the group element of $SU(2)$ can be parametrized as $S^3$ in the following way. Using the unit four-vector $a_i$, we have

$$U = a_0 \mathbb{I} + i\boldsymbol{a}\boldsymbol{\sigma}, \tag{116}$$

where $\sigma$ are the Pauli matrices, restricted to $\det U = a^2 = 1$. The Haar measure for $SU(2)$ is then

$$\mathrm{d}U = \frac{1}{\pi^2} \prod_{i=0}^{3} \mathrm{d}a_i \, \delta(a^2 - 1). \tag{117}$$

More generally, group elements of $SU(N)$ can be parametrized as

$$U(\omega) = \exp\left( i \sum_{j=1}^{N^2-1} \omega^{(j)} T_j \right), \tag{118}$$

where $\omega^{(j)}$, $j = 1, 2, \ldots, N^2 - 1$ are real numbers. It is easy to prove that $\left( \frac{\partial U(\omega)}{\partial \omega^{(k)}} U(\omega)^{-1} \right)$ is in the Lie algebra $\mathfrak{su}(N)$ of the group, so it makes sense to define the following metric

$$\mathrm{d}s^2 = g(\omega)_{nm} \, \mathrm{d}\omega^{(n)} \mathrm{d}\omega^{(m)}, \tag{119}$$

with

$$g(\omega)_{nm} = \mathrm{Tr}\left[ \frac{\partial U(\omega)}{\partial \omega^{(n)}} \frac{\partial U(\omega)^\dagger}{\partial \omega^{(m)}} \right]. \tag{120}$$

We can then define the measure $\mathrm{d}U$ as

$$\mathrm{d}g = c \sqrt{\det[g(\omega)]} \prod_k \mathrm{d}\omega^{(k)}, \tag{121}$$

where $c = 1/\int \prod_k \mathrm{d}\omega^{(k)} \sqrt{\det[g(\omega)]}$ guarantees the normalization condition $\int \mathrm{d}g = 1$. The invariance of such a measure can be easily seen from the change of the measure

$$\prod_k \mathrm{d}\omega^{(k)} = \det[J] \prod_k \mathrm{d}\omega'^{(k)} \tag{122}$$

under a transformation $\omega \mapsto \omega'$ with Jacobian $J_{ab} = \frac{\partial \omega^{(a)}}{\partial \omega'^{(b)}}$.

The invariance of the measure under left and right group multiplication automatically leads to

$$\int \mathrm{d}U \, U = \int \mathrm{d}U \, \Omega_1 U \Omega_2 = 0. \tag{123}$$

We are interested in computing observables $\mathcal{O}$, namely

$$\langle \mathcal{O} \rangle = \frac{1}{\mathcal{Z}} \int DU \, \mathcal{O}(U) e^{-S(U)}, \tag{124}$$

---

[6]As a result, in a lattice system the anomalies cannot arise from the lack of invariance of the path integral measure.

in a gauge-invariant theory. In the next subsection we will discuss observables that are of special interest for us.

An individual link variable, $U_\mu(x)$, is an example of a non-gauge invariant functional $\mathcal{O}(U)$, meaning $\mathcal{O}(U) \neq \mathcal{O}(U')$ for configurations $U$ and $U'$ related by a gauge transformation. On the other hand, the integration measure and the action, as the consequence of invariance under group multiplications and the gauge invariance of theory respectively, are gauge invariant. This immediately leads to the vanishing of not gauge invariant observables:

$$\langle \mathcal{O} \rangle = \frac{1}{\mathcal{Z}} \int DU e^{-S(U)} \mathcal{O}(U) = \frac{1}{\mathcal{Z}} \int DU' e^{-S(U')} \mathcal{O}(U) \neq \frac{1}{\mathcal{Z}} \int DU' e^{-S(U')} \mathcal{O}(U') = \langle \mathcal{O} \rangle \tag{125}$$
$$\Rightarrow \langle \mathcal{O} \rangle = 0 \,.$$

This is known as *Elitzur's theorem*. This in particular implies that a local order parameter such as $\langle U_\mu(x) \rangle$ is not useful for analyzing the phase structure of a gauge theory. In the next subsection we will discuss interesting gauge invariant observables.

## 4.2   Observables

In this subsection we discuss interesting observables, which are necessarily gauge invariant, of lattice gauge theories.

### 4.2.1   Wilson Loops

Wilson and Polyakov loop traces are essential observables, which can help us to determine the static quark-anti-quark potential, and provide valuable information about the confinement-deconfinement phase transitions. To discuss them, it would be necessary to distinguish between temporal and spatial directions. We hence denote a lattice point $x$ by $x = (\boldsymbol{x}, t)$, and in particular denote the edge Wilson line operator (93) as $U_\mu(x) = U_\mu(\boldsymbol{x}, t)$, in what follows.

Given a path $\mathcal{C}_{\boldsymbol{x}\boldsymbol{y}}$ connecting two points $\boldsymbol{x}$ and $\boldsymbol{y}$ on a constant time slice with time given by $t$, we consider the following two spatial Wilson lines $W(\boldsymbol{x}, \boldsymbol{y}, 0)$, $W(\boldsymbol{x}, \boldsymbol{y}, T)^\dagger$, where the spatial Wilson line connects two lattice points $\boldsymbol{x}, \boldsymbol{y}$ consisting only of spatial gauge links

$$W(\boldsymbol{x}, \boldsymbol{y}, T) := \prod_{((\boldsymbol{z}, T), \mu) \in \mathcal{C}_{\boldsymbol{x}\boldsymbol{y}}} U_\mu(\boldsymbol{z}, T) \,, \tag{126}$$

where we use $(z, \mu) = ((\boldsymbol{z}, T), \mu)$ to denote the oriented edge of the lattice starting at $z = (\boldsymbol{z}, T)$ and ending at $z + \hat{\mu} = (\boldsymbol{z} + \hat{\mu}, T)$, and the order of the product is taken following the path $\mathcal{C}_{\boldsymbol{x}\boldsymbol{y}}$. A completely analogous definition holds for $W(\boldsymbol{x}, \boldsymbol{y}, 0)$.

In addition, we consider two temporal lines $W(\boldsymbol{x}, T)^\dagger$ and $W(\boldsymbol{y}, T)$. The temporal Wilson line is a straight line of temporal gauge links located on a fixed spatial position

$$W(\boldsymbol{x}, T) = \prod_{j=0}^{T-1} U_0(\boldsymbol{x}, j) \,, \tag{127}$$

where $\mu = 0$ denotes the temporal direction, going from the past to the future.

Assembling the four Wilson lines together to form a closed loop $\mathcal{C}_x$ that starts and ends at the point $x = (\boldsymbol{x}, 0)$, the corresponding trace is a gauge invariant observable:

$$\mathcal{W}[\mathcal{C}_x] := \mathrm{Tr}\left(W(\boldsymbol{x}, \boldsymbol{y}, 0) W(\boldsymbol{y}, T) W(\boldsymbol{x}, \boldsymbol{y}, T)^\dagger W(\boldsymbol{x}, T)^\dagger\right) = \mathrm{Tr}\left(\prod_{(x, \mu) \in \mathcal{C}} U_\mu(x)\right) . \tag{128}$$

Notice that if we identify $U_\mu(x)$ with its exponential definition (94) we recover (92) in the continuous limit, which captures the monodromy of a quark traversing the contour $\mathcal{C}$.

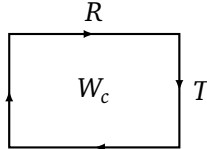

**Figure 11:** Wilson loop with two spatial and two temporal Wilson lines.

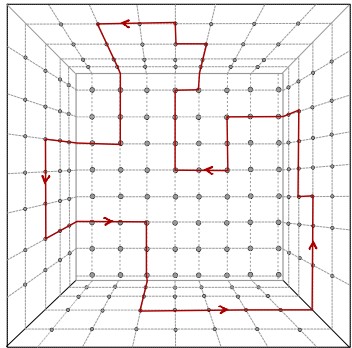

**Figure 12:** A Wilson loop on a 3d lattice.

An interpretation of the above Wilson loop is as the propagator of a state with two static charged particles at distance $R$, when $\mathcal{C}_{xy}$ is a straight path of length $R$. At large $T$ we hence expect

$$\lim_{T\to\infty}\langle\mathcal{W}[\mathcal{C}_x]\rangle \sim e^{-V(R)T}\,, \tag{129}$$

with $V(R)$ being the potential between static color charges, the so-called static quark-antiquark potential, which can be parametrized as

$$V(R) = A + \sigma R + \frac{B}{R}\,. \tag{130}$$

To see this, note that a simple counting in the strong coupling (small $\beta$) expansion of the action (98) leads to $V(R)\sim R$, while in the limit of weak coupling, the theory should behave like an Abelian theory with the usual Coulumb potential $V(R)\sim 1/R$.

As a result, when $\sigma \neq 0$ we have

$$\lim_{R\to\infty} V(R) \sim \sigma R\,, \tag{131}$$

and the Wilson loop at confinement follows an area law [66]

$$\lim_{R,T\to\infty}\langle\mathcal{W}[\mathcal{C}_x]\rangle \sim \exp\left(-\sigma R T\right)\,. \tag{132}$$

The parameter $\sigma$ is called the *string tension*; as the gluon field between quarks contracts to a tube or string, with energy proportional to its length. This is to be contrasted with a theory without confinement, in which the energy of a quark pair does not increase indefinitely with separation.

From the above, we see that the string tension can be extracted from the ratios [82]

$$\sigma = -\log\left(\frac{\langle\mathcal{W}(R,T)\rangle\langle\mathcal{W}(R+1,T+1)\rangle}{\langle\mathcal{W}(R+1,T)\rangle\langle\mathcal{W}(R,T+1)\rangle}\right) \tag{133}$$

and estimated in lattice simulations, where $\mathcal{W}(R,T)$ denotes the trace of a Wilson loop of the shape as shown in Figure 11.

### 4.2.2  Polyakov Loops

In §4.2.1 we have seen that at zero temperature, where the lattice extends infinitely in the time direction, the vacuum expectation value of the Wilson loop trace provides information about the static quark-antiquark potential. At finite temperatures, the Euclidean time direction is periodic, with periodic boundary condition for the gauge fields, and we instead consider the relation between the static quark-antiquark potential and the so-called Polyakov loops.

The Polyakov loop is a specific type of Wilson line that wraps the compactified (temporal) direction (from $t = 0$ to $t = N_t$). Explicitly, we have the following expression for the Polyakov loop at spatial location $\boldsymbol{x}$

$$P(\boldsymbol{x}) = \mathrm{Tr}\left[\prod_{j=0}^{N_t-1} U_0(\boldsymbol{x}, j)\right]. \tag{134}$$

The correlator of two Polyakov loops, depicted in Fig. 13, is given by the free energy of a quark-antiquark potential positioned at lattice sites $\boldsymbol{x}$ and $\boldsymbol{y}$, as

$$\langle P(\boldsymbol{x})P(\boldsymbol{y})^\dagger\rangle = e^{-aN_t V(R)}, \tag{135}$$

where $R = a|\boldsymbol{x} - \boldsymbol{y}|$ is the spatial distance between the two Polyakov loops as in (129).

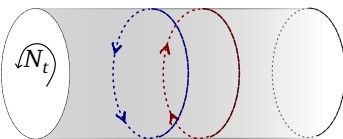

**Figure 13:** Polyakov loops of a quark and antiquark on a lattice with periodic time boundary conditions.

Now we turn to a single Polyakov loop, and consider

$$\langle P\rangle := \frac{1}{V}\sum_{\boldsymbol{x}}\langle P(\boldsymbol{x})\rangle.$$

It is related to the free energy of a single heavy quark via $\langle P\rangle \sim e^{-aN_t F_q}$.

At large distances, one expects

$$\lim_{|\boldsymbol{x}-\boldsymbol{y}|\to\infty}\langle P(\boldsymbol{x})P(\boldsymbol{y})^\dagger\rangle = \langle P(\boldsymbol{x})\rangle\langle P(\boldsymbol{y})^\dagger\rangle = |\langle P\rangle|^2. \tag{136}$$

For static potentials that exhibit infinite growth with increasing separation of the quarks, $|\langle P\rangle|$ must go to zero as a result of (135). Therefore, $\langle P\rangle$ must vanish in the confining case and can serve as an order parameter for the confinement-deconfinement phase transition:

$$\begin{aligned}\langle P\rangle = 0 &\Longleftrightarrow \text{ confinement },\\ \langle P\rangle \neq 0 &\Longleftrightarrow \text{ no confinement }.\end{aligned} \tag{137}$$

### 4.2.3  Spontaneous Breaking of the Center Symmetries

The above discussion about Polyakov loops can also be cast in the framework of symmetries. Here the relevant symmetry group is the center symmetry[7] $\mathfrak{C}_n \cong \mathbb{Z}/n$ of $SU(n)$, generated by the scalar multiplication by an $n$-th root of unity.

---

[7] The center is a subgroup that commutes with all other elements of the group: $Z(G) = \{z \in G : zg = gz \ \forall g \in G\}$.

In the lattice setup, the center transformation of $SU(3)$ gauge theory is defined as a scalar multiplication on all time-like gauge links originated at the spacetime points lying on a given temporal hyperplane: for fixed $t_0$ and any $x$, the transformation reads

$$U_0(\boldsymbol{x}, t_0) \mapsto z U_0(\boldsymbol{x}, t_0), \tag{138}$$

for a third root of unity. Explicitly, we can write

$$\mathfrak{C}_3 = \left\{ z\mathbb{1} : z = 1, e^{2\pi i/3}, e^{-2\pi i/3} \right\}. \tag{139}$$

To see that the action is invariant under the above transformation, note that the plaquette $P_{\mu,\nu}$ is left invariant as the center element commutes with all gauge links. However, it is easy to see that the Polyakov loop transforms as

$$P \mapsto z P, \tag{140}$$

as the closed loop go past the $t = t_0$ plane once.

In other words, when performing a center transformation, each quark line takes up a factor $z$ after winding around the temporal direction once. Correspondingly, an antiquark picks up a factor of $z^*$.

Note that the vaccum expectation value of the Polyakov loop must vanish when the center symmetry is unbroken, as

$$\langle P \rangle = \langle z P \rangle = \langle z^2 P \rangle = \langle P \rangle \times \frac{1}{3} \sum_{a \in \mathbb{Z}/3} e^{2\pi i a/3} = 0.$$

We hence conclude that, in a deconfined phase, where we encounter single quark states, the center symmetry must be broken as $\langle P \rangle \neq 0$ (cf. (137)).

### 4.2.4  Topological Charge

Another quantity in the continuous gauge theory in four dimensions of interest to us is the topological charge, a quantized quantity that characterizes different vacuum states and different topological sectors and is invariant under continuous deformations of the field. It is crucial to be able to measure the topological quantities meaningfully on a lattice, since topologically non-trivial configurations such as instantons play a vital role in understanding QCD; they are intrinsically related to the breaking of chiral symmetries, the axial anomalies, the strong CP problems, for instance.

Restricting to only field configurations with finite action, we require $F_{\mu\nu} \to 0$ at spatial infinity. One way to achieve this is by having $A_\mu = 0$. More generally, the field must be a pure gauge at spatial infinity:

$$A_\mu(x) \to \Lambda^{-1}(x) \partial_\mu \Lambda(x), \qquad |x| \to \infty. \tag{141}$$

As a result, we can specify the asymptotic limit of such a field configuration by specifying

$$\Lambda : S_\infty^3 \to SU(n). \tag{142}$$

Two such maps are said to be homotopic if they can be continuously deformed into each other. When $SU(2) \cong S^3$, it is intuitive that inequivalent maps $\Lambda$ have different winding numbers, counting how many times $S_\infty^3$ winds around $S^3 \cong SU(2)$. In general, equivalent maps $U$ are in the same homotopy class in the homotopy group $\pi_3(SU(n))$, which turns out to be the same for $n \geq 2$:

$$\pi_3(SU(n)) = \pi_3(SU(2)) = \mathbb{Z}.$$

One way to see this is from the fact that all topologically non-trivial $\Lambda$ (142), corresponding to the so-called large gauge transformations, can be deformed to lie within a single $SU(2)$.

Below we discuss how to compute the topological charge that gives us the above-mentioned winding number. From the field strength $F_{\mu\nu}$, one can build a natural four-index object (a "four-form") to integrate over the four-dimensional spacetime apart from the Yang-Mills action (88): one can show that

$$Q = \frac{1}{16\pi^2} \int d^4x \, \text{Tr}\left({}^\star F^{\mu\nu} F_{\mu\nu}\right) = \frac{1}{32\pi^2} \int d^4x \, \epsilon_{\mu\nu\rho\sigma} \text{Tr}(F^{\mu\nu} F^{\rho\sigma}) \in \mathbb{Z} \cong \pi_3(S^3). \quad (143)$$

To see how such a bulk four-dimensional integral can be related to the winding number which only depends on the field configuration at spatial infinity, we see that the integrand is in fact a total derivative (which is to say that the corresponding four-form is exact):

$$Q = \frac{1}{8\pi^2} \int d^4x \, \partial_\mu K^\mu, \quad K^\mu = \epsilon^{\mu\nu\rho\sigma} \text{Tr}\left(A_\nu \partial_\rho A_\sigma + \frac{2}{3} A_\nu A_\rho A_\sigma\right) \quad (144)$$

and the integral is therefore an integral over $S^3_\infty$ due to Gauss law. In particular, the following holds for the case of large gauge transformation (141):

$$K_\mu = -\frac{2}{3} \epsilon^{\mu\nu\rho\sigma} \text{Tr}\left[(\Lambda \partial_\nu \Lambda^{-1})(\Lambda \partial_\rho \Lambda^{-1})(\Lambda \partial_\sigma \Lambda^{-1})\right]. \quad (145)$$

leading to

$$Q = \frac{1}{24\pi^2} \int_{S^3_\infty} d^3x \, \epsilon^{\nu\rho\sigma} \text{Tr}\left((\Lambda \partial_\nu \Lambda^{-1})(\Lambda \partial_\rho \Lambda^{-1})(\Lambda \partial_\sigma \Lambda^{-1})\right). \quad (146)$$

This topological quantity measures the instanton charge of the field configuration. In terms of differential forms, one has the three-form given by $K = \text{Tr}\left(AdA + \frac{2}{3}A^3\right)$ and

$$Q = \frac{1}{8\pi^2} \int \text{Tr}(F \wedge F) = \frac{1}{8\pi^2} \int_{S^3_\infty} K. $$

### 4.2.5 Lattice Topological Charge Density

As mentioned in §2, there is strictly speaking no well-defined topological sectors in finite lattices. Nevertheless, it would be useful to define quantities that approach the topological charges in the limit of vanishing lattice spacing, as we aim to use finite lattices to approximate the physics of a continuous quantum field theory. Not surprisingly, the ways to achieve this are not unique. For a comparison of different definitions, see, for example [83].

**Gauge Theoretic Definition** A gluonic definition, depending only on the gauge fields, of the topological charge is given as the sum over the topological charge density

$$Q_L = a^4 \sum_{x \in V_L} q_x, \quad (147)$$

where

$$q_x = -\frac{1}{32\pi^2} \epsilon_{\mu\nu\rho\sigma} \text{tr}\left(\hat{F}_{\mu\nu}(x) \hat{F}_{\rho\sigma}(x)\right), \quad (148)$$

with $\hat{F}_{\mu\nu}$ being a discretization of the field strength tensor, given by for example (101). In practice, cooling/smearing is often applied to field configurations before measuring $Q_L$ via the

above formula, in order to suppress UV noises. While the instanton number $Q$ is an integer, the lattice counterpart generically does not satisfy this quantization property due to the effect of finite lattice spacing. Instead, it is possible to show that

$$Q_L = Q + \mathcal{O}(a^2), \quad \text{with} \quad Q \in \mathbb{Z}. \tag{149}$$

Specifically, UV fluctuations can cause spurious contributions to the topological charge, making it sometimes a complicated task to identify genuine instantons or other topologically significant structures.

To mitigate this problem by reducing the finite-size UV effects, one can improve the lattice discretization of the field strength tensor [84] using, for instance, the smearing techniques we introduced above[8].

**Fermionic Definition**    Another approach to define the topological charge is through considering the fermionic fields, via the Atiyah-Singer index theorem

$$\begin{aligned}\Delta Q_5 &= Q_5(\tau = \infty) - Q_5(\tau = -\infty) \\ &= \int \mathrm{d}^d x \int \mathrm{d}\tau\, \partial_0 J_0^5 = - \int \mathrm{d}^4 x\, \frac{N_F}{32\pi^2} \epsilon_{\mu\nu\rho\sigma} F^{\mu\nu} F^{\rho\sigma} = -2N_f Q,\end{aligned} \tag{150}$$

relating the topological charge to the difference between the number of left-handed and right-handed zero modes of the Dirac operator, with the factor given by the number of quark flavours $N_f$. This fermionic definition of topological charge is manifestly integer-valued and preserves the topological properties even on the lattice. However, it requires the use of a chiral-symmetric lattice Dirac operator, such as the overlap operator, which could be computationally demanding.

# 5    Fermions on a Lattice

In this section we will turn our attention to fermions, focussing on the theoretical aspects. For concreteness, we put some special emphasis on aspects relevant for the theory of QCD. We will briefly discuss the flow-based approaches to simulate lattice fermions in §6.3.

## 5.1    Discretizing Fermions

The QCD action can be split into two parts, $S_{\mathrm{QCD}} = S_{\mathrm{YM}} + S_F$. So far we have focused on the first, the Yang-Mills term. The second part of the action, $S_F$, involves both gauge fields and fermions, and is responsible for rich physics. In the context of the standard model, for instance, this part of the action describes the physics of the quarks. As such, in this section we will focus on 4-dimensions, when the representation theory of Clifford algebra requires us to specify the spacetime dimension in order to have a concrete discussion.

### 5.1.1    Free Fermions and Fermion Doubling

A naive discretization of the fermion action (86) leads to

$$S_F(\psi, \bar{\psi}, U) = a^d \sum_{x,y \in V_L} \bar{\psi}(x) \left[ \gamma^\mu \frac{1}{2a} (U_\mu(x)\delta_{x+a\hat{\mu},y} - U_{-\mu}(x)\delta_{x-a\hat{\mu},y}) + m\delta_{x,y} \right] \psi(y). \tag{151}$$

---

[8]Note however that the smearing can lead to errors caused by the fact that it could, potentially, favour classical solutions over time by disproportionately suppressing short-distance quantum fluctuations, i.e. after repeated application of smearing steps. As a result, care must be taken when choosing the smearing parameters.

To highlight the challenge of putting fermionic fields on a lattice in the simplest setting, in this section we will often ignore the interactions with the gauge fields. In the absence of the gauge fields ($U_\mu(x) = \mathbb{1}$), the fermion action (86) now reads

$$
\begin{aligned}
S_F^{\text{free}}(\psi, \bar{\psi}) &= a^d \sum_{x,y \in V_L} \bar{\psi}(x) \left[ \gamma^\mu \frac{1}{2a} (\delta_{x+a\hat{\mu},y} - \delta_{x-a\hat{\mu},y}) + m \delta_{x,y} \right] \psi(y) \\
&= a^d \sum_{x,y \in V_L} \bar{\psi}(x) D_F(x,y) \psi(y).
\end{aligned}
\tag{152}
$$

on a square lattice $V_L$. To examine the spectrum we write it in the momentum, using

$$
\delta_{x,y} = \int_{-\pi/a}^{\pi/a} \frac{\mathrm{d}^d x}{(2\pi)^d} e^{ip(x-y)},
\tag{153}
$$

where the range of integration are taken to be in the first Brillouin zone $(-\frac{\pi}{a}, \frac{\pi}{a}]$. Then, the Dirac matrix is given by

$$
D_F(p) = \gamma^\mu \frac{1}{2a} (e^{ip_\mu a} - e^{-ip_\mu a}) + m\mathbb{1} = \frac{i}{a} \gamma^\mu \sin(p_\mu a) + m\mathbb{1}.
\tag{154}
$$

Its inverse is then given by

$$
D_F^{-1}(p) = \frac{1}{\frac{i}{a} \gamma^\mu \sin(\pi_\mu a) + m\mathbb{1}} = \frac{-\frac{i}{a} \gamma^\mu \sin(p_\mu a) + m\mathbb{1}}{\frac{1}{a^2} \sin^2(p_\mu a) + m^2}.
\tag{155}
$$

In the limit $m \to 0$, one pole of the propagator is located at the expected location $p = (0,0,0,0)$ (assuming $d = 4$), consistent with the continuum theory. However there are $2^d - 1 = 15$ more poles at the other corners of the Brillouin zone $p \in \{ (\frac{\pi}{a},0,0,0), \ldots, (\frac{\pi}{a},\frac{\pi}{a},\frac{\pi}{a},\frac{\pi}{a}) \}$. These so-called 'doublers', which are purely artifacts of the lattice and have no analogs in the continuous theory, appear due to the perdiocity $p \sim p + 2\pi/a$ at finite $a$. Their presence is compatible with the lack of chiral anomalies in this naive lattice discretization. In particular, they correspond to eight zero modes with positive and eight with negative chirality [85], leading to cancelling contributions and thereby an anomaly-free theory.

Several approaches have been developed to address this problem of doublers and the chiral symmetries. We will discuss two such approaches in the what follows. The incompatibility between the doublers-free condition and exact chiral symmetries will be discussed in §5.3.1, and a way to get around it in §5.3.2.

### 5.1.2 Wilson Fermions

Wilson [86] proposed to introduce an additional term in the action that breaks chiral symmetries, but lifts the doubler states to higher masses, effectively removing them from low-energy physics.

$$
\begin{aligned}
S_F \to S_F^{(W)} &= S_F^{\text{free}} - a^d \frac{ra}{2} \sum_{x \in V_L} \overline{\psi}(x) \square \psi(x) \\
&= S_F^{\text{free}} - a^d \frac{r}{2a} \sum_{x,\hat{\mu}} \overline{\psi}(x) \{ \psi(x + a\hat{\mu}) + \psi(x - a\hat{\mu}) - 2\psi(x) \},
\end{aligned}
\tag{156}
$$

where $r$ is a parameter satisfying $0 < r \le 1$ and $S_F^{\text{free}}$ is given by (152). The lattice Laplacian operator is given by

$$
\square = \sum_{\hat{\mu}} \square_\mu, \quad \square_\mu f(x) = \frac{f(x + a\hat{\mu}) + f(x - a\hat{\mu}) - 2f(x)}{a^2}.
$$

The propagator now reads

$$D^{(W)}(p) = m\mathbb{1} + \frac{i}{a}\gamma^\mu \sin(p_\mu a) + \mathbb{1}\underbrace{\frac{r}{a}\sum_{\mu=1}^{d}\left(1 - \cos(p_\mu a)\right)}_{\text{Wilson term}}. \tag{157}$$

Note that the new 'Wilson' term vanishes at the continuum limit $a \to 0$. In the massless limit where $m = 0$, the zero at $p = (0,0,0,0)$ is intact while while other corners of the Brillouin zone $p \in \left\{\left(\frac{\pi}{a},0,0,0\right),\dots,\left(\frac{\pi}{a},\frac{\pi}{a},\frac{\pi}{a},\frac{\pi}{a}\right)\right\}$ acquire mass $2rl/a$, with $l$ the number of momentum components with $p_\mu = \pi/a$. As a result, there is only the light fermionic field as in the continuous limit and we have successfully elimited the fermions.

One issue of the Wilson term is the breaking of the chiral symmetry. Namely, even in the massless limit, we have

$$\left\{D^{(W)}(p), \gamma^5\right\} \neq 0,$$

since the Wilson term does not involve any $\gamma$ matrix. This will be discussed further in §5.3.

### 5.1.3   Staggered Fermions

In contrast to the above, the method of staggered fermions [87] does not seek to remove the fermion doublers directly. Instead, this method distributes these doublers on multiple lattice sites. At the end of the procedure, we are left with 4 'tastes', which are unphysical analogues of flavor, whose effects need to be removed afterward. The idea is to mix spacetime and spinor indices, which enables us to use Grassmanian fields with only one component (i.e. without a spinor index), and to subsequently reconstruct the physical multi-component spinor using the various poles of its propagator.

At the location $x = a(n_1, n_2, n_3, n_4), n_\mu \in \mathbb{Z}$ in the four-dimensional lattice, we introduce a new Dirac spinor $\psi'(x)$ by letting

$$\psi(x) = \gamma_1^{n_1}\gamma_2^{n_2}\gamma_3^{n_3}\gamma_4^{n_4}\psi'(x), \qquad \overline{\psi}(x) = \overline{\psi}'(x)\gamma_4^{n_4}\gamma_3^{n_3}\gamma_2^{n_2}\gamma_1^{n_1}. \tag{158}$$

Note that such a definition mixes the space-time and Dirac indices. Since $(\gamma^\mu)^2 = 1$ in the Euclidean space, the kinetic term of the discretized action will be given by

$$\gamma^\mu\psi(x \pm a\hat{\mu}) = \eta^\mu\gamma_1^{n_1}\gamma_2^{n_2}\gamma_3^{n_3}\gamma_4^{n_4}\psi'(x \pm a\hat{\mu}), \tag{159}$$

where the $\eta^\mu$ is the sign factor arising from commutating the $\gamma$-matrices:

$$\eta^\mu = (-1)^{\sum_{\nu < \mu} n_\nu}. \tag{160}$$

For example, we have

$$\eta_1(x) = 1, \quad \eta_2(x) = (-1)^{n_1}, \quad \eta_3(x) = (-1)^{n_1+n_2}, \quad \eta_4(x) = (-1)^{n_1+n_2+n_3}, \tag{161}$$

again for $x = a(n_1, n_2, n_3, n_4)$. For later use, we also define $\eta_5(x) = (-1)^{n_1+n_2+n_3+n_4}$. Importantly, in terms of $\psi'$ the action becomes diagonal in Dirac indices

$$\overline{\psi}(x)\gamma_\mu\psi(x \pm a\hat{\mu}) = \eta^\mu\overline{\psi}'(x)\mathbb{1}\psi'(x \pm a\hat{\mu}). \tag{162}$$

As the right-hand side of the above equation decomposes into identical equations for each spinor component, we are free to retain only one component and discard the rest. By doing so, we can reduce the number of the unwanted fermionic doublers. Denote by $\chi(x)$ a Grassman field with

only color indices and no Dirac structure, starting from (151), the resulting staggered fermionic action is given by

$$S_F(\chi,\bar{\chi},U) = a^4 \sum_{x\in V_L} \bar{\chi}(x)\left(\sum_{\mu=0}^{3} \eta^\mu(x)\frac{U_\mu(x)\chi(x+\hat{\mu}) - U_{-\mu}(x)\chi(x-\hat{\mu})}{2a} + m\chi(x)\right). \quad (163)$$

Compared to the action of usual fermions, the additional, location-dependent factor $\eta^\mu(x)$ prevents the appearance of additional zero modes arising from the corners of the lattice Brillouin zone.

**Tastes**   To recover the spinor structure in staggered fermions, recall that we have reduced the number of degrees of freedom by mixing the spinor and spacetime indices. We will therefore reconstruct the spinor structure by combining $\chi$-fields at different spacetime locations. To see how this works, first note that, due to the phase $\eta^\mu(x)$, the staggered action has translational invariance only under shift by an integral multiple of $2a$. Thus, one can argue that, in the continuum limit, a $2^4$ hypercube is mapped to a single point.

In view of the above, we write $b = 2a$ and the location $x = a(n_1,n_2,n_3,n_4), n_\mu \in \mathbb{Z}$ in the four-dimensional lattice as

$$x = b(h_1,h_2,h_3,h_4) + a(s_1,s_2,s_3,s_4), \quad h_\mu \in \mathbb{Z}, \quad s_\mu \in \{0,1\}$$

and treat the $s_\mu$ indices as "internal" indices. These $2^4 = 16$ internal degrees of freedom can be thought of of as four sets, or four "tastes" of four Dirac components. To see the spinor structure, we define $s$-dependent matrices $\Gamma^{(s)}$:

$$\Gamma^{(s)} = \gamma_1^{s_1}\gamma_2^{s_2}\gamma_3^{s_3}\gamma_4^{s_4} \quad (164)$$

and assemble the new quark fields as

$$q(h)_{\alpha\beta} \equiv \frac{1}{8}\sum_{s\in\{0,1\}^4} \Gamma^{(s)}_{\alpha\beta}\chi(2h+s), \quad \bar{q}(h)_{\alpha\beta} \equiv \frac{1}{8}\sum_{s\in\{0,1\}^4} \bar{\chi}(2h+s)\Gamma^{(s)*}_{\alpha\beta}, \quad (165)$$

where $\alpha,\beta = 1,\ldots,4$. Interpreting the first index as the Dirac index and the second index as the taste label, $\psi^{(t)}(h)_\alpha := q(h)_{\alpha t}$, we obtain the following action for free ($U_\mu(x) = 1$) staggered fermions: [88, 89]

$$S^{(\text{stag})} = b^4 \sum_h \left( \sum_t \left(m\bar{\psi}^{(t)}(h)\psi^{(t)}(h) + \bar{\psi}^{(t)}(h)\gamma^\mu\nabla_\mu\psi^{(t)}(h)\right) \right.$$
$$\left. -\frac{b}{2}\sum_{t,t'}(\tau_5\tau_\mu)_{tt'}\bar{\psi}^{(t)}(h)\gamma_5\Box_\mu\psi^{(t')}(h) \right), \quad (166)$$

where $b = 2a$ and $\tau_\mu = \gamma_\mu^T$. In the above, we have the lattice derivatives

$$\nabla_\mu f(h) = \frac{f(h+b\hat{\mu}) - f(h-b\hat{\mu})}{2b}, \quad \Box_\mu f(h) = \frac{f(h+b\hat{\mu}) + f(h-b\hat{\mu}) - 2f(h)}{b^2}$$

as usual. Note that the sum over $h$ should be interpreted as the sum over the lattice where the lattice spacing is twice the spacing of the original lattice, $b = 2a$. While the first two terms of the action are diagonal in the taste labels, the last term mixes between the different tastes.

**Symmetries** One of the appealing features of staggered fermions is that they preserve a subset of the chiral symmetries. To see this, note that in the massless chiral limit, the action (163) is invariant under the $U(1)$ transformation

$$\chi(x) \mapsto e^{i\alpha\eta_5(n)}\chi(x), \quad \bar{\chi}(x) \mapsto e^{i\alpha\eta_5(n)}\bar{\chi}(x).$$

**Staggered Rooting** In simulations, the staggered fermions with the action (163) have the advantage in terms of computational efficiency by having fewer degrees of freedom. From performing the path integral over the Grassmannian field $\psi, \bar{\psi}$.

$$\mathcal{Z} = \int \prod_{x,\mu} \mathrm{d}U_\mu(x) \prod_x [\mathrm{d}\bar{\psi}_x\, \mathrm{d}\psi_x] e^{-S_G(U)-\bar{\psi}D(U)\psi} = \int \prod_{x,\mu} \mathrm{d}U_\mu(x) \det[D(U)] e^{-S_G(U)},$$

we propose to simulate the fermions using the following effective action involving the fourth-root of the determinant of the staggered fermions:

$$e^{-S_{\text{eff}}(U)} = e^{-S_G(U)} \left(\det\left[D^{(\text{stag})}(U)\right]\right)^{1/4}. \tag{167}$$

Despite their advantage in computational efficiency, conceptual puzzles remain for the staggered fermions. For instance, the rooting procedure leads to undesirable non-locality. See [90, 91, 92]. That said, the numerical results obtained using staggered fermions appear to be in good agreement with experimental results.

## 5.2 Chiral Symmetries

In this section we discuss chiral symmetries of the massless fermion field theory, before putting the theory on the lattice. In QCD, chiral symmetries arise in the limit of vanishing quark masses. Although the measured quark masses are not zero, the masses of the two lightest quarks, up and down, are very small compared to hadronic scales. Therefore, chiral symmetry can be regarded as an approximate symmetry of the strong interactions, explicitly broken by the quark masses. However, it can also be spontaneously broken, giving rise to approximately massless Goldstone bosons (e.g. pions). Effective models incorporating these light mesonic degrees of freedom can describe the low-energy behavior of QCD. However, a first-principle understanding of the precise details of the underlying dynamics responsible for these mechanisms of chiral symmetry breaking remains unclear. The realization of chiral symmetries on a lattice will be discussed in the next subsection.

Recall that the fermionic fields can be decomposed into the right- and left-handed fermion fields

$$\psi_R = P_R\psi, \qquad \psi_L = P_L\psi, \qquad \bar{\psi}_R = \bar{\psi}P_L, \qquad \bar{\psi}_L = \bar{\psi}P_R \tag{168}$$

using the projection operators

$$P_L = \frac{1}{2}(\mathbb{1} + \gamma_5), \qquad P_R = \frac{1}{2}(\mathbb{1} - \gamma_5), \quad \text{with} \quad P_R + P_L = \mathbb{1} \quad \text{and} \quad P_R P_L = P_L P_R = 0. \tag{169}$$

With these, the Lagrangian for a theory of $N_f$ quark flavors

$$\mathcal{L} = \bar{\psi}(\slashed{D} + M)\psi, \quad \text{where} \quad \slashed{D} = \gamma^\mu(\partial_\mu + iA_\mu) \tag{170}$$

and $M = \mathrm{diag}(m_1, m_2, \dots m_{N_f})$, can be written as

$$\mathcal{L} = \bar{\psi}_R \slashed{D} \psi_R + \bar{\psi}_L \slashed{D} \psi_L + M(\bar{\psi}_R \psi_L + \bar{\psi}_L \psi_R), \tag{171}$$

using the anti-commutation relation

$$\{\slashed{D}, \gamma_5\} = 0. \tag{172}$$

In the absence of the masses, one can easily see that the left- and right-handed components completely decouple, contributing independently to the Lagrangian.

Therefore the massless Lagrangian is invariant under independent unitary transformations

$$\psi_{L,R} \mapsto U_{L,R}\psi_{L,R}, \tag{173}$$

captured by the symmetry group $U(N_f)_L \times U(N_f)_R$, which we decompose into

$$SU(N_f)_V \times SU(N_f)_A \times U(1)_V \times U(1)_A. \tag{174}$$

The transformation of the $U(1)_V$ symmetry group is given by

$$U(1)_V : \psi' = e^{i\alpha\mathbb{1}}\psi, \quad \bar{\psi}' = \bar{\psi}e^{-i\alpha\mathbb{1}}, \quad \alpha \in \mathbb{R} \quad \text{and} \quad \mathbb{1} = \mathbb{1}_{N_f}, \tag{175}$$

leading to the conservation of the quark number, and similarly for $SU(N_f)_V$, with $\mathbb{1}_{N_f}$ replaced by generators of $SU(N_f)$. Finally, the chiral or axial vector rotations are defined as

$$U(1)_A : \psi' = e^{i\alpha\gamma_5\mathbb{1}}\psi, \qquad \bar{\psi}' = \bar{\psi}e^{i\alpha\gamma_5\mathbb{1}}, \tag{176}$$

$$SU(N_f)_A : \psi' = e^{i\alpha\gamma_5 T_i}\psi, \qquad \bar{\psi}' = \bar{\psi}e^{i\alpha\gamma_5 T_i}, \tag{177}$$

where $T_i$ are the generators of $SU(N_f)$. The $U(1)_A$ symmetries in eq. (176) are explicitly broken by the non-invariance of the fermion due to the *axial anomaly*, namely the invariance of the integration measure under the transformations. The $SU(N_f)_A$ symmetries in eq. (177) are part of the symmetry group $SU(N_f)_L \times SU(N_f)_R$ and are spontaneously broken, by the formation of non-zero condensates of up quarks

$$\langle \bar{u}(x)u(x) \rangle, \tag{178}$$

leading to the remaining symmetries $SU_R(N_f) \otimes SU_L(N_f) \to SU_V(N_f)$. In what follows we will discuss these breakings of chiral symmetries in more details.

### 5.2.1   Axial Anomaly

As mentioned above, although the Lagrangian is invariant under the transformations of eq. (176), $U(1)_A$ is not a true symmetry of QCD due to the non-trivial topological structure the QCD vacuum [93]. Noether's theorem states that global symmetries lead to conserved currents, and in the case of the axial symmetry, the current reads

$$J_\mu^5 = \bar{\psi}\gamma_\mu\gamma_5\psi. \tag{179}$$

While classically conserved, $\partial^\mu J_\mu^5 = 0$, the symmetry is broken at the quantum level by the chiral anomaly, giving rise to [94, 95]

$$\partial^\mu J_\mu^5(x) = \frac{N_f}{32\pi^2}\epsilon_{\mu\nu\rho\sigma}\text{tr}(F^{\mu\nu}F^{\rho\sigma}). \tag{180}$$

This result can be derived using Feynman diagrams calculations, where the right-hand side can be attributed to one-loop diagrams. Alternatively, it can be derived through a regularized computation of the change of fermionic integration measure under an axial transformation, to which only the fermionic zero modes contribute. The relation between the two points of view is the essence of the Atiyah-Singer index theorem [96]. See [97], for instance, for a detailed discussion.

In more details, recall that the pseudoscalar density entering the anomaly is a total divergence, as shown in (144). From the discussion in §4.2.5, we see that $A_\mu$ is generically a non-trivial pure gauge field at spatial infinity, implying that at the $U(1)_A$ symmetry is broken at the quantum level. As a result, there is hence no associated Goldstone boson. Specifically, the change in the axial charge is proportional to the topological charge (143)

$$\Delta Q_5 = Q_5(\tau = \infty) - Q_5(\tau = -\infty)$$
$$= \int \mathrm{d}^d x \int \mathrm{d}\tau \, \partial_0 J_0^5 = \int \mathrm{d}^4 x \, \frac{N_f}{16\pi^2} \epsilon_{\mu\nu\rho\sigma} \mathrm{tr}(F^{\mu\nu} F^{\rho\sigma}) = 2N_f Q \,, \tag{181}$$

where

$$Q = \int \mathrm{d}^4 x \, q(x) = \int \mathrm{d}^4 x \, \frac{1}{32\pi^2} \epsilon_{\mu\nu\rho\sigma} \mathrm{tr}(F^{\mu\nu} F^{\rho\sigma}) \,. \tag{182}$$

As alluded to above, this axial anomaly can also be derived from the counting of fermionic zero-modes. Consider the massless Dirac operator $\slashed{D} = \gamma_\mu(\partial_\mu + A_\mu)$. From the anti-commutation (172) we see that it is possible to attribute definite handedness ($\gamma_5 \psi = \pm \psi$) to the zero modes satisfying $\slashed{D}\psi = 0$. The theorem then states that the difference of the left and right handed zero modes is given by the topological charge

$$\mathrm{index}(\slashed{D}) = n_+ - n_- = 2N_f Q \,. \tag{183}$$

This result shows that, interestingly, the axial anomalies arise due to the non-trivial topology of the gauge field configurations, and in particular the instanton-like configurations with $Q \neq 0$.

This result holds significant implications for the axial anomaly: instanton-like configurations, with $Q \neq 0$, in the vacuum of the theory, interact with fermions, flipping the chirality of some of their zero-modes. Overall, the chiral anomaly can be understood as a consequence of the topology of the background field. This anomaly reduces the full $U(N)_A \simeq SU(N)_A \otimes U(1)_A$ symmetry that is present in the classical theory to $SU(N)_A$ which is then spontaneously broken, as we will discuss shortly.

The realization that topologically non-trivial configurations make a genuine contribution to the dynamics introduced a new challenge in QCD. Note that the factor $\epsilon_{\mu\nu\rho\sigma} F^{\mu\nu} F^{\rho\sigma}$, or equivalently $F \wedge F$, breaks the CP symmetry as it is invariant under charge conjugation, but not under parity or time reversal transformations. In the absence of the CP symmeties, such a term is allowed in the Lagrangian density, involving an additional free parameter known as the vacuum angle $\theta$. So we should consider

$$\mathcal{L}_\theta = -i\theta q(x) \,. \tag{184}$$

Generically, we also expect such a term to arise from RG transformations. A first consequence is that the vacuum state receives contributions from configurations with all topological charges. It is a superposition across different topological sectors characterized by a winding number $n$: $|\theta\rangle = \sum_{n=-\infty}^{\infty} e^{-in\theta} |n\rangle$. The puzzle, often referred to as the strong CP problem, is why the $\theta$ vacuum angle is measured to be below $10^{-10}$ [98], and hence is considered unnaturally small and "fine tuned". A number of possible explanations have been proposed, including postulating the existence of a new particle, the axion (cf. [99] for a review) which would be a pseudo-Nambu-Goldstone boson for a new hypothetical global $U(1)$ symmetry (Peccei-Quinn symmetry) that is spontaneously broken.

The topological nature of the $\theta$ term has significant physical consequences: the vacuum state of the theory cannot be constructed as a quantum fluctuation around a classical well-defined state

[100]. Thus, a non-perturbative approach is essential to thoroughly investigate the physics of the $\theta$ parameter. In principle this could be studied by the lattice regularization similar to the discussions in the previous sections. However, a non-vanishing $\theta$ angle leads to the following sign (or phase) problem when trying to simulate the lattice field theories.

**Sign Problem**   To see this, note that the introduction of the $\theta$ term (184) introduces a phase $e^{i\theta Q[A]}$ in the weight $e^{-S}$ when evaluated for a topologically non-trivial field configuration $A$, obstructing the interpretation of $e^{-S}/\mathcal{Z}$ as giving a density distribution in the space of field configurations. A straightforward approach to this problem involves incorporating $\exp(-\operatorname{Im}S[\phi])$ into the observable and then sampling exclusively with $\exp(-\operatorname{Re}S[\phi])$ a strategy known as reweighting. Namely,

$$\langle \mathcal{O}_S \rangle = \frac{\langle \exp(-\operatorname{Im}S[\phi])\,\mathcal{O}[\phi] \rangle_{\operatorname{Re}S}}{\langle \exp(-\operatorname{Im}S[\phi]) \rangle_{\operatorname{Re}S}}. \tag{185}$$

However, even in the case in which $\operatorname{Re}S$ does not vanish, the performance of this procedure is doomed to perform poorly since

$$\langle \exp(-\operatorname{Im}S[\phi]) \rangle_{\operatorname{Re}S} = \frac{\int D\phi \exp(-S[\phi])}{\int D\phi \exp(-\operatorname{Re}S[\phi])} = \frac{\mathcal{Z}}{\mathcal{Z}_R} \tag{186}$$

and

$$|\mathcal{Z}| = \left| \int D\phi\, e^{-S} \right| \leq \int D\phi \left| e^{-S} \right| = \int D\phi\, e^{-ReS} = \mathcal{Z}_R. \tag{187}$$

As $\mathcal{Z}$ and $\mathcal{Z}_R$ are both extensive quantities, we see that there exists a constant $c > 0$ such that

$$\langle \exp(-\operatorname{Im}S[\phi]) \rangle_{\operatorname{Re}S} \sim e^{-cV} \tag{188}$$

and the number of configurations needed increases exponentially with the system size $V$ due to the large cancellations between contributions from different configurations with different phases. In fact, the sign problem has been proven to be NP-hard [101], making the existence of an effective algorithm highly improbable. Nevertheless, algorithms including the complex Langevin method [102, 103], extending the fields to the complex plane and sampling from a positive-definite distribution, and Lefschetz thimbles methods [104] which deforms the original integration contour in the complexified field space to a new set of contours where the imaginary part of the action is constant, have been proposed to combat the phase problem.

### 5.2.2   Spontaneous Chiral Symmetry Breaking

A symmetry of the action is considered spontaneously broken if the ground state does not preserve it. In the case of continuous groups, this spontaneous symmetry breaking leads to the emergence of massless particles known as Goldstone bosons. Recall that the symmetries in our case, $SU_A(N_f)$, is an approximate symmetry and is only realized exactly in the limit of vanishing quark masses. As we mentioned, this approximate symmetry is broken by the quark condensate (178). It is easy to see that the chiral condensate breaks the $SU(N_f)_A$ symmetries by noting that it behaves as a mass term. Its expectation value serves as an order parameter for chiral symmetry breaking, vanishing in the chirally symmetric phase, and non-zero in the broken phase.

In the chiral limit, the presence of an octet of the lightest pseudoscalar mesons can be understood in the light of the spontaneous breaking of $SU(N_f)_A$, as the 8 Goldstone bosons parametrizing the broken symmetry group $SU(N_f)$, which would have been strictly massless in the case of exactly massless fermions and acquire small masses in the QCD case of approximate chiral symmetries. In the real world, where quark masses are small but non-zero and there

was no exact chiral symmetry to start with. The consequence is that the would-be Goldstone bosons – the pions, kaons and the eta mesons– acquire small masses. Nevertheless, the fact that these particles are much lighter than typical hadrons can be attributed to the approximate symmetry that gets spontaneously broken, and demonstrates that spontaneous symmetry breaking is a useful concept even in the case of approximate symmetries.

Another way to see the breaking of chiral symmetries manifests itself is through the absence of parity doublets in nature. In the massless limit, the Lagrangian (171) is symmetric under the chiral symmetries in (174). Since axial transformations mix states of opposite parity, the symmetry implies that for each hadron state, there should be a partner with opposite parity and (approximately) equal mass. However, this is not observed in nature at low energies, indicating that the symmetry is 'spontaneously broken'. For instance, consider the nucleon ground state $N(940)$ and the $N(1535)$ resonance with the opposite parity, and note that they have very distinct masses, 940MeV and 1535MeV respectively.

If chiral symmetry were to be unbroken, the two states would be degenerate. Instead, the large mass gap we observed suggests that the vacuum state of QCD does not respect chiral symmetry.

Finally, lattice QCD simulations provide additional evidence for spontaneous chiral symmetry breaking. At low temperatures, numerical studies confirm the presence of a non-zero chiral condensate, in agreement with theoretical predictions [105, 106, 107]. At high temperatures (above the crossover temperature that is around $150 - 160$ MeV) the condensate vanishes, indicating that the chiral symmetry is restored in a quark-gluon plasma phase.

## 5.3   Chiral Symmetries on a Lattice

When discretizing space-time on a lattice, maintaining both chiral symmetry and locality turns out to be problematic, as the Nielsen-Ninomiya theorem states. This can be expected from the following heuristic argument. Recall that the lattice QCD has finitely many degrees of freedom and is hence free from anomalies, while the anomalies are present in the continuum limit, we see that to avoid contradictions, the lattice action cannot maintain naive chiral symmetries.

This has direct implications for numerical lattice simulations, as implementing chiral symmetries often requires computationally intensive methods. Wilson fermions introduce explicit symmetry breaking, requiring fine-tuning, while Ginsparg-Wilson fermions (e.g., overlap fermions) preserve chiral symmetry but are computationally demanding due to their non-local nature. Consequently, the complexities of chiral symmetry present challenges to achieving efficient and accurate lattice QCD simulations.

### 5.3.1   The Nielsen-Ninomiya Theorem

In §5.1 we see that a naive discretization of the fermionic field theory leads to the problem of fermion doubling. It was later realized that the fermion doublers cannot be removed in a way that is compatible with chiral symmetries [108]. The formal statement is the following:

> **Theorem 1: Nielsen-Ninomiya No-Go Theorem**
> There does not exist a lattice Dirac operator $D_{\text{lat}}$ which satisfies all of the following conditions
>   1. **Correct limit**: $\lim_{a \to 0} D_{\text{lat}} = \slashed{D}$ .
>   2. **Doubler-free** Additional fermion species decouple in continuum limit.
>   3. **Locality**: $\exists r, c, A \geq 0$ s.t. $|D_{\text{lat}}(x, y)| < A e^{-c|x-y|}$.

4. **Chiral symmetry:** $\{D_{\text{lat}}, \gamma^5\} = 0$ for $m = 0$.

A schematic proof is given in [109]. Examples of this no-go theorem can be found in §5.1. Heuristically, one recalls that on a $d$-dimensional lattice Brillouin zone exhibits the topology of a torus, $T^d$, on which one would like to define the Dirac-like operator. The periodicity of the Brillouin zone implies that the the operator generally possesses multiple zero modes. Second, for such an operator that anticommutes with $\gamma_5$, each zero mode can be assigned a definite chirality. The index theorem on the compact torus dictates that the chiralities of the zero-modes must be paired, and leading to the impossibility of removing all doublers.

We have seen that a naive Dirac-like operator satisfying the conditions $1, 3, 4$ necessarily breaks $2$. Next, we will proceed by giving up $4$ instead.

### 5.3.2   The Ginsparg-Wilson Relation

Given the Nielsen-Ninomiya No-Go Theorem discussed above, we will now proceed by modifying the meaning of chiral symmetry on the lattice. The modification is given by the Ginsparg-Wilson relation [110]

$$\{D, \gamma^5\} = aD\gamma^5 D\,, \tag{189}$$

satisfied by a lattice Dirac operator $D$, which reduces to the anticommutation relation eq. (172) in the continuum $a \to 0$ limit. In the full notation of eq. (151), we write

$$\gamma_5 D(x, y) + D(x, y)\gamma_5 = a \sum_z D(x, z)\gamma_5 D(z, y)\,. \tag{190}$$

This relation implies that the action is not invariant under the chiral transformation $\delta\psi = i\theta\gamma_5\psi$:

$$\delta(\bar{\psi}D\psi) = (\delta\bar{\psi})D\psi + \bar{\psi}D\delta\psi = i\theta\bar{\psi}(\gamma_5 D + D\gamma_5)\psi = i\theta\bar{\psi}(aD\gamma_5 D)\psi \neq 0\,. \tag{191}$$

Instead, a chiral rotation leaving the lattice action, with the lattice Dirac operator satisying the Ginsparg-Wilson relation, is given by [111]

$$\psi \to \psi' = e^{i\theta\gamma_5(1-\frac{a}{2}D)}\psi\,, \quad \bar{\psi} \to \bar{\psi}' = \bar{\psi}e^{i\theta(1-\frac{a}{2}D)\gamma_5}\,, \tag{192}$$

which recovers the usual chiral transformation as $a \to 0$. The Lagrangian density for massless fermions is left invariant under this transformation:

$$
\begin{aligned}
\mathcal{L}[\psi', \bar{\psi}'] &= \bar{\psi}'D\psi' \\
&= \bar{\psi}\exp\left(ia\left(1 - \frac{a}{2}D\right)\gamma_5\right)D\exp\left(ia\gamma_5\left(1 - \frac{a}{2}D\right)\right)\psi \\
&= \bar{\psi}\exp\left(ia\left(1 - \frac{a}{2}D\right)\gamma_5\right)\exp\left(-ia\left(1 - \frac{a}{2}D\right)\gamma_5\right)D\psi \\
&= \bar{\psi}D\psi = \mathcal{L}[\psi, \bar{\psi}]\,,
\end{aligned}
\tag{193}
$$

where we use the Ginsparg-Wilson equation in disguise

$$D\gamma_5\left(1 - \frac{a}{2}D\right) + \left(1 - \frac{a}{2}D\right)\gamma_5 D = 0\,. \tag{194}$$

The lattice projection operators are then given by

$$\hat{P}_R = \frac{1 + \hat{\gamma}_5}{2}\,, \quad \hat{P}_L = \frac{1 - \hat{\gamma}_5}{2}\,, \quad \hat{\gamma}_5 = \gamma_5(1 - aD)\,, \tag{195}$$

where due to the Ginsparg-Wilson relation, $\hat{\gamma}_5^2 = 1$, implying the usual properties for the projection operators, including

$$\hat{P}_R \hat{P}_L = \hat{P}_L \hat{P}_R = 0, \quad \hat{P}_L + \hat{P}_R = \mathbb{1}. \tag{196}$$

Notice that the continuum relation $\gamma^\mu P_{L/R} = P_{R/L} \gamma^\mu$, which is a consequence of the continuum chiral symmetry $\{\not{D}, \gamma^5\} = 0$ and which leads to the decoupling of left-moving and right-moving massless fermions, now becomes

$$D\hat{P}_R = P_L D, \quad D\hat{P}_L = P_R D, \tag{197}$$

to correctly reflect the lattice version of the chiral symmetry. It is then sensible to define

$$\psi_R = \hat{P}_R \psi, \quad \psi_L = \hat{P}_L \psi, \quad \bar{\psi}_R = \bar{\psi} P_R, \quad \bar{\psi}_L = \bar{\psi} P_L, \tag{198}$$

so that the mixed terms $\bar{\psi}_L D \psi_R$ and $\bar{\psi}_R D \psi_L$ vanish, and the action decouples to left and right handed parts, like in the continuum:

$$\bar{\psi} D \psi = \bar{\psi}_L D \psi_L + \bar{\psi}_R D \psi_R. \tag{199}$$

A crucial difference between the continuum case and the lattice one is that in the continuum, chiral rotation and projection only involves the spinors at a given point $x$. On the lattice, it is required the application of the lattice Dirac operator $D$. That is chirality of a lattice fermion involves information of neighboring sites and depends on the gauge field.

### 5.3.3 The Overlap Operator

Coming back to the Nielsen-Ninomiya theorem, we have relaxed the fourth condition on the chiral symmetry, and now we would like to make sure that a candidate lattice Dirac operator satisfies the rest of the conditions, in particular the absence of the doublers and locality. One such candidate is the overlap operator [112, 113]:

$$D_{\text{ov}} = \frac{1}{a} \left( \mathbb{1} + \gamma_5 \text{sgn}[H] \right). \tag{200}$$

In the above, we choose $H = \gamma_5 h$, with $h$ some suitable "kernel" Dirac operator that is $\gamma_5$-Hermitian. That is, $\gamma_5 h \gamma_5 = h^\dagger$ so $H$ is Hermitian with real eigenvalues. The matrix

$$\text{sgn}[H] = H(H^\dagger H)^{-1/2} \tag{201}$$

has the same set of eigenvectors as $H$, but with their eigenvalues replaced by $\pm 1$, the sign of the eigenvalues of $H$. This "overlap operator" satisfies the Ginsparg-Wilson relation, as can be shown from the short calculation below.

**Proof:**

$$
\begin{aligned}
a D_{\text{ov}} \gamma_5 D_{\text{ov}} &= a \frac{1}{a} (1 + \gamma_5 \text{sgn}[H]) \gamma_5 \frac{1}{a} (1 + \gamma_5 \text{sgn}[H]) \\
&= \frac{1}{a} (\gamma_5 + \gamma_5 \text{sgn}[H] \gamma_5 + \gamma_5 \gamma_5 \text{sgn}[H] + \gamma_5 \text{sgn}[H] \gamma_5 \gamma_5 \text{sgn}[H]) \\
&= \frac{1}{a} (1 + \gamma_5 \text{sgn}[H]) \gamma_5 + \gamma_5 \frac{1}{a} (1 + \gamma_5 \text{sgn}[H]) \\
&= D_{\text{ov}} \gamma_5 + \gamma_5 D_{\text{ov}} \\
&= \{D_{\text{ov}}, \gamma_5\}.
\end{aligned}
\tag{202}
$$

The kernel is often chosen to be doubler-free Dirac-Wilson operator (157)

$$H = \gamma_5 D^{(W)}. \tag{203}$$

The main computational challenge in calculating the overlap Dirac operator is to evaluate $\text{sgn}[H]$, which is given by the orthonormal eigenvectors and eigenvalues of $H$ as

$$\text{sgn}[H] = \sum_i \text{sgn}(\lambda_i) \, |i\rangle \langle i| \,. \tag{204}$$

However, a direct computation through diagonalization is infeasible due to the large size of $H$. Instead, the expression (201) can be approximated via the Zolotarev rational polynomial approximation of $(H^2)^{-1/2}$:

$$\text{sgn}(H) \approx d \frac{H}{\lambda_{\min}} \left( \frac{H^2}{\lambda_{min}^2} + c_{2n} \right) \sum_{l=1}^{n} \frac{b_l}{\frac{H^2}{\lambda_{\min}^2} + c_{2l-1}}, \tag{205}$$

where the coefficients are given in [114] or [88].

## 5.4   Dynamical Fermions

Formally, the lattice partition function for a system of fermions and gauge fields is

$$\mathcal{Z} = \int \prod_{x,\mu} dU_\mu(x) \prod_x [d\bar{\psi}_x \, d\psi_x] e^{-S_G(U) - \bar{\psi}_x D(U) \psi_x}, \tag{206}$$

where $S_G(U)$ is the gauge action, and $\bar{\psi} D(U) \psi$ is the fermion action. Because of its quadratic form, the integration over the Grassmann fields can be carried out analytically, leading to

$$\mathcal{Z} = \int \prod_{x,\mu} dU_\mu(x) \det[D(U)] e^{-S_G(U)}. \tag{207}$$

This fermionic factor can be treated in different ways, depending on the amount of simplifications one is willing to accept and the physics one wishes to study. Dynamical fermions play a crucial role in lattice gauge theories, where they represent fermions that interact dynamically with the gauge fields rather than being treated as static background particles. Unlike in quenched approximations, where fermion effects are either ignored or overly simplified, dynamical fermions actively contribute to vacuum fluctuations, playing a significant role in phenomena such as chiral symmetry breaking and confinement in QCD.

### 5.4.1   Pseudofermions

A direct calculation of the fermionic determinant $\det[D[U]]$ in (207) is typically prohibitively expensive. Specifically, the fermionic action $S_F[\psi_x, \bar{\psi}_x, U] = \bar{\psi}_x D[U] \psi_x$ leads to the Grassmanian integral [9]

$$\int D[\psi_x, \bar{\psi}_x] e^{-S_F[\psi_x, \bar{\psi}_x, U]} = \det D[U] = \prod_{f=1}^{N_f} \det D_f[U]. \tag{208}$$

---

[9] A Grassmannian variable $\psi$ satisfy the integration rules

$$\int D\psi = 0, \quad \int D\psi \, \psi = 1.$$

Note that the matrix $D[U]$ has color, flavour, Dirac, and lattice indices. In particular, the size of the Dirac matrices $D_f$ scales as the total number of lattice sites, rendering the calculation very costly. As a result, one might attempt to evaluate the fermionic determinant as an observable depending on $U$. However, depending on the gauge configuration $U$, the Dirac operator can have a widely different set of eigenvalues, leading to a difficult sampling problem where an accurate estimattion requires an extremely large number of samples due to the large variance.

A common strategy is to calculate it using pseudofermion fields – complex bosonic fields $\phi$ that reproduce the effects of fermionic determinants, using the identity

$$\frac{1}{\det A} = \frac{1}{\mathcal{Z}_N} \int D\phi\, e^{-\phi^\dagger A \phi}, \quad \text{with} \quad \mathcal{Z}_N = \int D\phi\, e^{-\phi^\dagger \phi}, \tag{209}$$

which holds for any matrix $A$ that has all eigenvalues having positive real parts.

However, the Dirac matrices $D_f$ are typically not positive-definite, which prevents one from applying this identity to each factor $\det D_f$. In the case of degenerate fermion pairs, such as up and down quarks (i.e. $N_f = 2$) in the usual approximation, their pair Dirac operators are equal $D_1 = D_2 = D$, which allows us to combine their determinants as

$$\det D \det D = \det\left(DD^\dagger\right), \tag{210}$$

which is manifestly positive-definite for which (209) applies. In the above, we have used the $\gamma_5-$Hermiticity of the Dirac operator, $D^\dagger = \gamma_5 D \gamma_5$, which implies

$$\det\left(D^\dagger\right) = \det(\gamma_5 D \gamma_5) = \det D.$$

Applying the above, instead of sampling from the probability density

$$p(U) = \frac{1}{\mathcal{Z}} e^{-S_G(U)} \det D[U], \tag{211}$$

which incurs computational costs that scale poorly with the system size, we can sample from the joint distribution

$$p(U, \phi) = \frac{1}{\mathcal{Z}'} e^{-S_G(U) - S_{\text{pf}}(\phi)} = \frac{1}{\mathcal{Z}'} e^{-S_{\text{eff}}}, \quad \text{with} \quad S_{\text{pf}} = \phi^\dagger (DD^\dagger)^{-1} \phi, \tag{212}$$

with any variant of Hybrid Monte Carlo algorithms, for example cf. [115], or recently, with exact generative sampling schemes, as explored in [37, 116, 117], which will be discussed in §6.3. Note that the effective action is very non-local, as it involves the inverse of the Dirac operator.

The challenge of evaluating the determinant of the operator is replaced by the challenge of inverting the Wilson-Dirac operator. An indicator for difficulty of the inversion is the *condition number* $\kappa(D) = \frac{\lambda_{max}}{\lambda_{min}}$, where a small condition number (close to 1) means that the matrix is well-conditioned, making numerical inversion stable and efficient, whereas a large condition number means that the matrix is ill-conditioned, making numerical solutions unstable and slow to converge. The Wilson-Dirac operator falls in the latter category, since when approaching the chiral limit and the quark masses go to zero, the smallest eigenvalues become small. To address this problem, *preconditioning* techniques are used, which modify the condition number, making the inversion faster and more stable. See for instance [118].

# 6  Simulating Gauge Theories Using Normalizing Flows

In this section, we combine the elements reviewed in the previous sections, and discuss how normalizing flows can be employed to assist sampling in lattice gauge theories. One of the most interesting challenges in designing such normalizing flows is equivariance. While we have discussed the case of equivariance under global symmetries (cf. section 3.5), applications in gauge theories require equivariance under the spacetime-dependent gauge symmetries. Below we discuss different approaches to equivariance including spectral flows, lattice-CNN [119], and Lie algebraic transformations. When combined with different flow architectures such as the coupling layers of Real NVP [35, 45] and continuous flows via neural ODEs [44, 43], we can build equivariant normalizing flows that are potentially powerful tools for lattice gauge theory simulations. In view of the applications we have in mind, we will restrict our discussions to special unitary gauge groups $G = SU(N)$.

## 6.1  Gauge Equivariant Transformations

In what follows we will consider equivariance with respect to the following two types of transformations, which are relevant for Wilson loops and for link variables respectively. Namely, as follows from (91) and (92), under a gauge transformation given by $\{\Omega(x) \in G\}_{x \in V_L}$, a link variable transforms as

$$\Omega : U_\mu(x) \mapsto (\Omega \cdot U)_\mu(x) := \Omega(x) U_\mu(x) \Omega^{-1}(x + \hat{\mu}). \tag{213}$$

Therefore, a gauge equivariant transformation $f_L : G \to G$ on gauge links should satisfy

$$f_L \cdot \Omega = \Omega \cdot f_L. \tag{214}$$

Following the transformation of gauge links, a Wilson loop $W[\mathcal{C}_x]$ starting and ending at point $x$, and in particular the plaquette operator $P_{\mu,\nu}(x)$, transforms as a conjugation (cf. (89))

$$\Omega : W[\mathcal{C}_x] \mapsto \mathrm{Conj}_{\Omega(x)}(W[\mathcal{C}_x]). \tag{215}$$

For such Wilson loop operators, the link equivariance (214) implies that the resulting map $f_W$ on loop operators should transform like

$$f_W \cdot \mathrm{Conj}_\Omega = \mathrm{Conj}_\Omega \cdot f_W. \tag{216}$$

### 6.1.1  Spectral Flow

For the sake of discussion, in this part we first consider a single Wilson loop $W[\mathcal{C}_x]$, for instance the plaquette $P_{\mu,\nu}(x)$, and describe a "spectral" transformation that is equivariant under conjugation, as required by the Wilson loop gauge equivariance (215).

Since one can always diagonalize a given group element $U$ by conjugation, for a map $f$ satisfying the equivariance condition (216), one only needs to specify its action on diagonal elements

$$\Lambda = \mathrm{diag}\left(e^{i\theta_1}, \ldots, e^{i\theta_{N-1}}, e^{-i(\theta_1 + \cdots + \theta_{N-1})}\right).$$

In other words, we can consider the action of $f$ on the maximal torus $T \subset G$ of the gauge group, and then extend it to the whole group via equivariance [10]. See Appendix A of [45] for details.

---

[10]A maximal torus $T$ in $G = SU(N)$ is a maximal abelian subgroup consisting of all diagonal elements. Note that

The equivariance (216) in particular implies equivariance with respect to the Weyl group, or the permutation of eigenvalues. One way to construct such a map is to simply consider $f$ preserving a fundamental (or canonical) cell of the maximal torus under the action of the Weyl group, and then extend it using the Weyl group to the whole maximal torus.

Based on these principles, a spectral flow was constructed in [120, 45] as a permutation equivariant flow on the fundamental cell of the maximal torus. The conjugation-equivariant transformation of an arbitrary element $U \in SU(N)$ can then be given by the following three-step process :

1. Diagonalize the matrix $W$ into $W = V\Lambda V^{-1}$, with a choice of $V$ such that

$$\Lambda = \mathrm{diag}\left(e^{i\theta_1}, \ldots, e^{i\theta_{N-1}}, e^{-i(\theta_1 + \cdots + \theta_{N-1})}\right)$$

   is in the fundamental cell,

2. Apply a permutation-equivariant transformation $\Lambda \mapsto \Lambda'$, which can be parametrized by a neural network,

3. Obtain the transformed matrix $W'$ via $W' = V\Lambda'V^{-1}$.

### 6.1.2   Lie Algebraic Transformations

Next we turn our attention to constructing diffeomorphisms on $G$ that are equivariant with respect to the link transformation (213). It is convenient to parametrize such a map as

$$U \mapsto g(U) \cdot U,$$

where the multiplication is element-wise in the link space. When the above transformation is thought of as a small step making up a long sequence of transformations, it is convenient to parametrize $g(U)$ in terms of Lie algebra elements:

$$U \mapsto e^{iX(U)}U, \quad \text{with } X(U) \in \mathfrak{g} . \tag{217}$$

For instance, the stout transformation (108) is naturally of this form.

For the transformation to be equivariant with respect to the gauge transformation (213), we need the component of $g(U)$ that corresponds to the link starting from the point $x$ to be transformed by a conjugation[11]

$$g_{x,\mu}(\Omega \cdot U) = \mathrm{Conj}_{\Omega(x)} g_{x,\mu}(U). \tag{218}$$

This is the case if the Lie algebra element $X_{x,\mu}(U)$ transforms according to the corresponding adjoint action $\mathrm{Ad} : G \to \mathrm{Aut}(\mathfrak{g})$:

$$X_{x,\mu}(\Omega \cdot U) = \mathrm{Ad}_{\Omega(x)} X_{x,\mu}(U). \tag{219}$$

In §6.2.2 and §6.2.3, we will discuss in more details the ways to realize such a gauge equivariant transformation.

the maximal torus is equal to its own centralizer

$$T \cong Z(T) := \{X \in G | XDX^{-1} = D, \forall D \in T\} .$$

The Weyl group of $G$ is the group of symmetries on the maximal torus $T$ that preserves the torus: $W := N(T)/T$, where the normalizer subgroup is defined as

$$N(T) = \{X \in G | XDX^{-1} \in T, \forall D \in T\} .$$

Here, we use the fact that, acting on $T$, it is isomorphic to the group of permutations on the $N$ entries on the diagonal.

[11] The asymmetry between the end points arise due to the fact that we mainly consider left group multiplication in our calculation. Alternatively, an analogous setup can be built using right group multiplication.

### 6.1.3  L-CNN

Lattice Gauge Equivariant Convolutional Neural Networks (L-CNNs), introduced in [119], are a modification of conventional CNNs designed to handle the symmetries of lattice gauge theories. In particular, L-CNNs are manifestly equivariant under gauge symmetries.

In L-CNNs, data points are represented as tuples $(\mathcal{U}, \mathcal{W})$, where $\mathcal{U} = \{U_\mu(x)\}_{\mu,x}$ denotes the gauge links of a particular lattice configuration, and $\mathcal{W} = \{W_i(x)\}_{i,x}$ is a set of Lie group elements, associated to each lattice point $x$ and transforming under gauge transformation via conjugation by the local gauge transformation attached to $x$, just as untraced Wilson loops starting and ending at the point $x$ do (cf. (215)). The authors of [119] proposed to consider the following types of transformations on such a tuple $(\mathcal{U}, \mathcal{W})$.

**L-Convolutions**    A L-convolution transofmration maps $(\mathcal{U}, \mathcal{W}) \mapsto (\mathcal{U}, \mathcal{W}')$ via

$$W_i'(x) = \sum_{j,\mu,k} \omega_{ij\mu k} U_{k\mu}(x) W_j(x + k\mu) U_{k\mu}^\dagger(x), \tag{220}$$

where $\omega_{ij\mu k} \in \mathbb{C}$ are the trainable convolution weights, $i$ and $j$ are the output and input indices respectively, $\mu \in \{0, \dots, D-1\}$ denote the lattice directions and the distance is given by $k \in \{-K, -K+1, \dots, K\}$. In the above, $U_{k\mu}(x)$ denotes the product of link variables in the $\mu$ (or $-\mu$ if $k < 0$) direction forming a straight path from $x$ to $x + k\mu$. In this sense, $K \in \mathbb{N}$ determines the kernel size, or the size of the convolution window.

For a given $k\mu$, the corresponding summand in (220) has the effect of extending $W(x + k\mu)$ with paths connecting it to the point $x$, making it transform like an object associated to the point $x$ and thereby preserving the gauge transformation (215).

**L-Bilinear layer**    The L-Bilinear layer, which consists of local maps of the form

$$W_i(x)' = \sum_{jk} a_{ijk} W_j(x) W_k(x), \tag{221}$$

is bilinear in $W_i(x)$ and does not mix features that are associated with different points $x \neq y$, where $a_{ijk} \in \mathbb{C}$ are taken to be trainable weights. Together with L-Convolutions, it is capable of modelling non-local and non-linear transformations of the Lie group-valued features $\mathcal{W} = \{W_i(x)\}_{i,x}$. It manifestly preserves the gauge transformation (215).

By stacking these layers, one can show that all arbitrary contractible Wilson loops can be constructed in this way, which together with the topological information such as contractible Wilson loop traces gives the full information of the gauge connections on a lattice [119]. Additional layers such as gauge equivariant activation layers can also be included, and together they can be used to build a gauge-equivariant neural network, which can be incorporated into different architectures. We will discuss some of them in the next subsection. For instance, when the goal is to produce useful gauge invariant features, one can simply take the trace at the end of the chain of transformations and obtain

$$\mathcal{T}_{x,i}(\mathcal{U}, \mathcal{W}) = \mathrm{Tr}[W_i(x)], \tag{222}$$

which is gauge invariant by virtue of (215). The equivariance property makes L-CNN a potentially useful tool for constructing gauge equivariance flows. For instance, one might imagine using L-CNN to parametrize the gauge invariant functions parametrizing the Lie algebra flow discussed in §6.2.2 and §6.2.3.

## 6.2 Flow Architectures

In this subsection we discuss how the gauge-equivariant transformations can be used as building blocks to construct gauge equivariant normalizing flows that can be applied to lattice gauge theory simulations. The normalizing flows can then be trained by minimizing the loss function (59), given by the reverse KL divergence. In particular, for a flow map $f$ to perfectly "normalize" the physical theory with density given by $p(U) = e^{-S(U)}/\mathcal{Z}$, namely, for the inverse map $f^{-1}$ to map $p(U)$ to the Haar measure, it must satisfy (52)

$$S(f(U)) - \log|\det J_f(U)| = \text{constant}. \tag{223}$$

### 6.2.1 Gauge Equivariant Real NVP

In [35, 120], it was proposed to adopt the structure of coupling layers, reviewed in §3.3, to the case of lattice gauge theories. In particular, we devise a coupling layer mapping the link variables $G^{|E|} \rightarrow G^{|E|}$, by first dividing the links into two subsets, the frozen subset $\mathcal{U}^A$ and the active subset $\mathcal{U}^B$, and devising the transformations on the plaquettes accordingly. We will use a specific masking pattern, shown in Figure 15, as an example to explain the idea. More discussions on the masking pattern can be found in [120]. In order to update the links $U_\mu(x)$ (in green in Fig 15), we do so by transforming the plaquette $P_{\mu,\nu}(x)$ (in orange in Fig 15) to ("active update")

$$P'_{\mu,\nu}(x) = h(P_{\mu,\nu}(x); I), \tag{224}$$

where $h(P_{\mu,\nu}(x); I)$ is a gauge-equivariant neural network satisfying the conjugation equivariance (216), for instance given by a spectral flow reviewed in §6.1.1. As reviewed in §3.3, the idea is to ensure that the contextual information about the existing configuration is included in the map, and to have a tractable Jacobian simultaneously, by letting the context functions $I = \{I_\alpha\}_\alpha$ depend on the frozen links $U_{\mu'}(x') \in \mathcal{U}^A$ alone. A convenient choice to design such equivariant maps $h$ is to require all features $I_\alpha$ to be gauge invariant.

After transforming the plaquette $P_{\mu,\nu}(x)$, the next step is to transform the link $U_\mu(x)$ accordingly. To do this, we assume that the change of plaquette $P_{\mu,\nu}(x) \mapsto P'_{\mu,\nu}(x)$ is solely due to the change of the link $U_\mu(x)$, while the other three links in $P_{\mu,\nu}(x)$ (95) are held fixed. This leads to the following transformation depicted in Figure 14.

$$U'_\mu(x) = P'_{\mu,\nu}(x)P^\dagger_{\mu,\nu}(x)U_\mu(x). \tag{225}$$

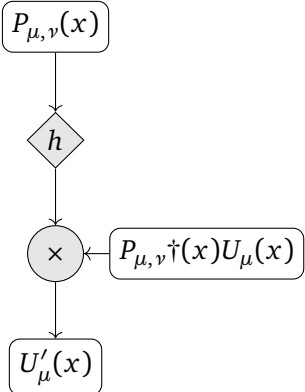

**Figure 14:** Transformation of $U_\mu(x)$ link.

Note that, by updating the link $U_\mu(x)$, the other plaquette which contains $U_\mu(x)$ also gets updated passively ("passive updates"), as shown in Figure 15.

It is clear that, depending on the exact masking pattern, the whole set of link variables requires a few coupling layers to be fully updated. Relatedly, due to the masking structure, the architecture is not equivariant with respect to the whole group of spatial shift symmetries of the lattice; it is broken to a smaller subgroup when compared to case of scalar field theories. See §3.3. This constitutes one of the motivations to look at continuous flows, which we will discuss shortly.

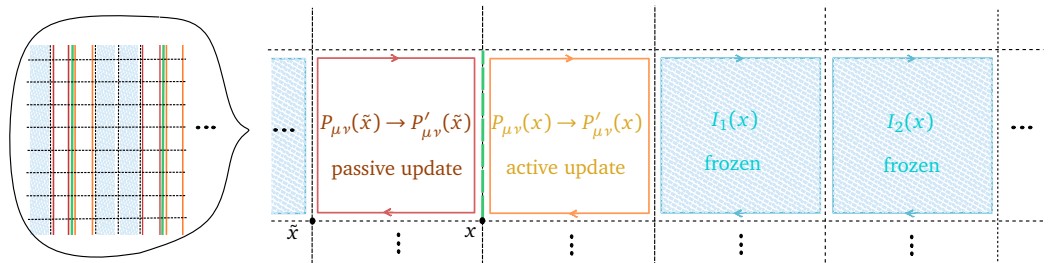

**Figure 15:** Single coupling layer updating a subset of gauge links (in green).

### 6.2.2  Gauge Equivariant ResNet

Gauge equivariant ResNet is another architecture for gauge equivariant flows which has been applied to lattice gauge theories in works including [121, 120, 122].

The architecure of an i-ResNet given by (63) can be generalized to Lie groups by considering the residual blocks, which transforms the gauge links with a Lie algebraic transformation (217). By stacking such transformations

$$U^{(i+1)} = e^{iX^{(i)}(U^{(i)})}U^{(i)}, \tag{226}$$

which are given by Lie algebra elements $X^{(i)}(U)$ as discussed in §6.1.2, we can build an expressive transformation which is invertible as long as the appropriate Lipschitz conditions are satified.

Various methods in §6.1 can be applied here to construct the Lie algebra element $X^{(i)}$ satisfying the equivariance condition (219). First, note that the stout smearing transformation (109) has the desired equivariance condition (219), since both $Q_\mu(x)$ and $Q_\mu^\dagger(x)$ both transforms according to (218). Hence, one way to construct Lie algebra element $X^{(i)}$ is to generalize the stout transformation, by separating the gauge links into the active and frozen ones as in Real NVP, and by iteratively transforming the Wilson-loop type features, in each step depending only on the frozen gauge links, from which one can build the Lie algebra element by a projection similar to (109). Another method is by taking derivatives of gauge invariant quantities. Consider $X(U)$ given by

$$X(U) = \sum_a T_a \partial_a \tilde{S}(U), \tag{227}$$

where

$$\partial_a \tilde{S}(U) := \frac{\mathrm{d}}{\mathrm{d}s} \tilde{S}\big(e^{isT_a}U\big)\big|_{s=0}. \tag{228}$$

When $S$ is a gauge invariant function of the gauge links, the Lie algebra element $X(U)$ transforms under gauge transformation in the desired way (219).

A more formal way to understand the transformation (217)-(227), which will help us to understand the gauge equivariance, is the following. Note that $d\tilde{S} \in \mathfrak{g}^*$, with $d\tilde{S} : \mathfrak{g} \to \mathbb{C}$ by $d\tilde{S}(u) = \sum_a u^a \partial_a \tilde{S}(U)$, where we write $u \in \mathfrak{g}$ in the basis $\{T_a\}$ as $u = \sum_a u^a T_a$. Using the invariant bilinear form $\langle,\rangle : \mathfrak{g} \times \mathfrak{g} \to \mathbb{C}$, which is non-degenerate, unique and in particular equal

to the Killing form to a scalar factor [12], we can identify $\mathfrak{g} \cong \mathfrak{g}^*$ and associate to $d\tilde{S}$ an element $X(U) := \sum_a T_a \partial_a \tilde{S}(U)$ of the Lie algebra $\mathfrak{g}$. Recall that the conjugation transformation (89) leads to the adjoint action $\mathrm{Ad} : G \to \mathrm{Aut}(\mathfrak{g})$ on Lie algebra by taking derivatives at the origin $e \in G$, which leaves the Killing form invariant. As $\tilde{S}(U)$ is gauge invariant, it is easy to see that under a gauge transformation (213), we have

$$\partial_a \tilde{S}(U) = \partial_{a'} \tilde{S}(\Omega \cdot U), \quad \text{with } T'_a := \mathrm{Ad}_{\Omega(x)} T_a \tag{229}$$

and the Lie algebra element $X(U)$ hence transforms as

$$X(\Omega \cdot U) = \sum_a T_a \partial_a \tilde{S}(\Omega \cdot U) = \sum_{a'} T'_a \partial_{a'} \tilde{S}(\Omega \cdot U) = \sum_{a'} T_{a'} \partial_a \tilde{S}(U) = \mathrm{Ad}_{\Omega(x)}(X(U)) \tag{230}$$

and this proves the equivariance of the Lie algebraic transformation (217) with the above construction of the Lie algebra element.

In the above we have suppressed the link indices, given by $(x, \mu)$ as in (93), to avoid cluttering. In the lattice gauge theory application, we consider $\tilde{S}(U)$ that depends on all gauge links $U_\nu(x')$ with all directions $\nu$ and sites $x'$. We therefore define

$$e^{isT_a^{(x,\mu)}} U_\nu(x') = \begin{cases} e^{isT_a} U_\mu(x) & \text{if } (x, \mu) = (y, \nu), \\ U_\nu(x') & \text{otherwise}, \end{cases}$$

with which we define

$$X_\mu(x) = \sum_a T_a \partial_a^{(x,\mu)} \tilde{S}(U), \tag{231}$$

where

$$\partial_a^{(x,\mu)} \tilde{S}(U) := \frac{\mathrm{d}}{\mathrm{d}s} \tilde{S}\left(e^{isT_a^{(x,\mu)}} U\right)\Big|_{s=0}. \tag{232}$$

### 6.2.3  Gauge Equivariant Continuous Flows

As mentioned in §3.4.2, neural ODE provides a natural infinitesimal, continuous version of the ResNet architecture featured in the above discussion. Moreover, neural ODE has the advantage of being a flexible architecture which can accommodate a lot of expressivity and can incorporate equivariance in an elegant way. Below we will discuss the application of neural ODE to gauge theories.

To discuss the general structure of the gauge equivariant continuous flows, first note that an infinitesimal version of the Lie algebraic transformation (217) is given by

$$\dot{U}(t) = Z(U, t) U(t), \tag{233}$$

where in a neural ODE $Z_\theta : G^{|E|} \times [0, T] \to \mathfrak{g}^{|E|}$. As before, we will often write the Lie algebra element as $Z^{(x,\mu)}(U, t) = \sum_a T_a Z^{a,(x,\mu)}(U, t)$ in terms of the Lie algebra basis, where we have reintroduced the link label $(x, \mu)$. In the case of such an ODE map $f_t : G^{|E|} \to G^{|E|}$, the change of the density is again given by the divergence of the vector field (77), which for the Lie group flow (233) reads

$$\frac{\mathrm{d}}{\mathrm{d}t} \log p_t(U(t)) = -\frac{\mathrm{d}}{\mathrm{d}t} \mathrm{tr}\left(J_{f_t}(U)\right) = -\nabla \cdot Z(t, U) = -\sum_a \sum_{(x,\mu) \in E} \partial_a^{(x,\mu)} Z^{a,(x,\mu)}(t, U). \tag{234}$$

---

[12]for instance, it can be chosen to be the trace of products in the fundamental representation, $\langle x, y \rangle = -\frac{1}{2} \mathrm{Tr}(\rho_F(x)\rho_F(y))$.

**Trivializing Flows** In the context of lattice gauge theories, the proposal of "trivializing flows" was in fact proposed before the advent of flow-based generative models [123]. The idea is to utilize an ODE transformation that "trivializes" the theory by mapping the physical density to the Haar measure (223), which leads to easy sampling with the Hamiltonian Monte Carlo method.

First, he noted that one way to solve the trivializing condition (223) with the ODE flow (233) satisfying (234) is to require the vector field to satisfy

$$\int_0^t ds \, \nabla \cdot Z(s, U(s)) = tS(U(t)) + c(t), \tag{235}$$

for some field independent function $c(t)$. This corresponds to choosing a particular path that connects the Haar measure ($t = 0$) and the physical distribution ($t = 1$) in the space of distributions on $G^{|E|}$.

With the same argument as presented in §6.2.2, we see that the ODE map (233) has the desired equivariance property (214) if the Lie algebra element $Z(U, t)$ is given by derivatives of a gauge invariant function:

$$Z^{a,(x\mu)}(U, t) = \partial_a^{(x,\mu)} \tilde{S}(U, t). \tag{236}$$

With this Ansatz, the flow condition (235) becomes

$$\mathfrak{L}_t \tilde{S}(t, U) = S(U) + \dot{c}(t)$$
$$\mathfrak{L}_t = -\Box + t \nabla S \cdot \nabla = \sum_a \sum_{(x,\mu) \in E} -\partial_a^{(x,\mu)} \partial_a^{(x,\mu)} + t(\partial_a^{(x,\mu)} S) \partial_a^{(x,\mu)}. \tag{237}$$

Moreover, the Lüscher flow further assumes that the gauge invariant potential $S(U, t)$ admits a polynomial expansion in the flow time $t$: $\tilde{S}(U, t) = \sum_{k \geq 0} t^k \tilde{S}_k(U)$. With this assumption, the above equation is solved recursively with

$$\tilde{S}_0 = \Box^{-1} S, \quad \tilde{S}_k = -\Box^{-1} \sum_a \sum_{(x,\mu) \in E} \partial_a^{(x,\mu)} S \partial_a^{(x,\mu)} \tilde{S}_{k-1}. \tag{238}$$

In the case of pure gauge Wilson action (100), the action is proportional to the inverse temperature $\beta$, and the above recursive relation shows that the perturbation in ODE time of the potential $\tilde{S}$ is simultaneously an expansion in $\beta$, with $\tilde{S}_k \sim \beta^{k+1}$, and is only applicable at strong coupling. This is one of the reasons why the beautiful idea of Lüscher did not immediately lead to great numerical results. A machine-learning enhanced implementation of a trivializing flow was recently reported in [44] which showed promising results.

**Non-Perturbative Trivializing Flows** Tapping more into the general expressive power of neural networks, below we will discuss a more general continuous flow, which does not depend on such a perturbative expansion [43].

First we note that the Ansatz (236) for $Z(U, t)$ can clearly be generalized to the case with more than one gauge invariant functions, with coefficients which are themselves gauge invariant functions, allowing for more expressivity and without impacting the gauge equivariance property of the map. Namely, we can consider

$$Z_\theta(U, t) = \sum_a T_a Z_\theta^a(U, t) = \sum_\alpha \Lambda_\alpha(U, t) \sum_a T_a \partial_a \tilde{S}_\alpha(U), \tag{239}$$

where $S_\alpha(U)$ and $\Lambda_\alpha(U, t)$ are all gauge invariant functions of the gauge links, which can be parametrized by a neural network.

To build these gauge invariant functions, one can take as fundamental building blocks the traces of various Wilson loops. To create a sufficiently expressive ordinary differential equation for the flow, we need to include a larger variety of Wilson loops beyond just the plaquettes. One way of arranging these Wilson loops is by choosing an anchor point $\bar{x}$, which we take to be a vertex of the dual graph, and organize the Wilson loops we utilize via their class $k$, given the anchor point $\bar{x}$. As an example, fig. 16 illustrates the classes of loops considered in [43] (up to a rotation of the loop), with the black dots indicating the anchors. Denoting the trace of the corresponding Wilson loop operators by $\mathcal{W}^{(k)}(\bar{x})$, in [43] the following parametrization of the Lie algebra was considered

$$Z^{a,(x,\mu)}(t, U) = \sum_{k,\bar{x}} \left( \partial_a^{(x,\mu)} \mathcal{W}^{(k)}(\bar{x}) \right) \Lambda_{\bar{x}}^{(k)}(t, \mathcal{W}), \tag{240}$$

with $\Lambda(t, \mathcal{W})$ a time-dependent convolution of features obtained from $\mathcal{W}$ by locally applying a neural network at each anchor point $\bar{y}$:

$$\Lambda_{\bar{x}}^{(k)} = \hat{\Lambda}^{(k)}(t) + \sum_{j,\bar{y}} C_{\bar{x}\bar{y}}^{kj}(t)\, g_{\bar{y}}^{(j)}(t, \mathcal{W}(\bar{y})), \tag{241}$$

where $C(t)$ is a time-dependent convolution kernel, $\hat{\Lambda}^{(k)}(t)$ are field-independent learnable parameters, and $g$ represents a neural network.

When choosing the set of Wilson loops to include, it is advisable to take into account the so-called Mandelstam constraints [124, 125, 126], derived from identities satisfied by the traces of $SU(N)$ matrices and giving rise to linear dependencies among traces of different Wilson loops. For instance, for any $SU(3)$ matrix $U$ one has

$$(\mathrm{Tr}\, U)^2 = \mathrm{Tr}(U^2) + 2\, \mathrm{Tr}\, U^\dagger. \tag{242}$$

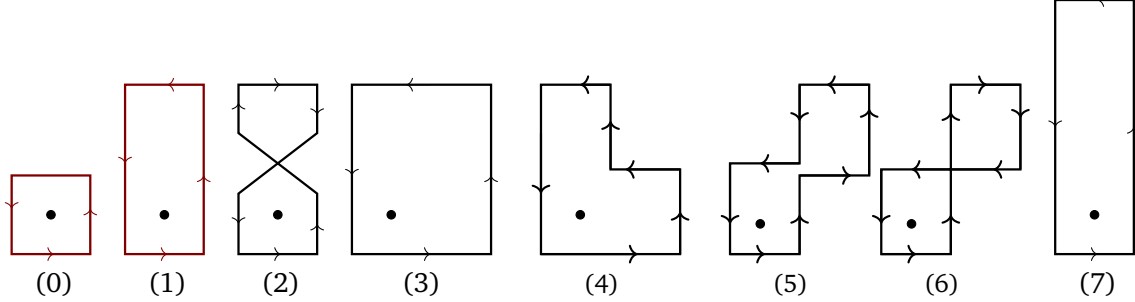

**Figure 16:** Example of classes of loops.

As mentioned in §3.4.2, neural ODE allows for computation of the final density and the loss gradient via solving additional ODEs (77) and (74), which in the gauge theory case can be achieved using the Crouch-Grossman Runge-Kutta integration schemes reviewed in appendix A. In this case, the evolution of the probability density is now given by (234). To compute the density, it is crucial that the Ansatz (240) allows for an efficient computation of the divergence of the vector field.

In the case of the loss gradient, one needs to generalize the adjoint method to allow for general manifolds, in particular Lie groups. We refer the readers to [43] for more details. For a general

flow on a manifold $\mathcal{M}$ given by the ODE $\dot{z} = f_\theta(z)$, $z \in \mathcal{M}$, the gradient of the loss $L(z(T))$ given a fixed initial value $z(0)$ can be computed using

$$d_\theta L(z(T)) = -\int_T^0 a(t) \circ d_\theta f_\theta(t, z(t)) \, dt \, , \tag{243}$$

by integrating backwards from $T$. This uses the auxiliary *adjoint state* $a(t)$, which is formally a cotangent vector of $\mathcal{M}$. It is initialized as $a(T) = d_z L\big|_{z(T)}$ and solved for simultaneously using

$$\dot{a}(t) = -a(t) \circ d_z f_\theta\big|_{z(t)} \, . \tag{244}$$

Here, we have used the more abstract notation from differential geometry. In particular, for $\varphi : \mathcal{M} \to \mathcal{M}'$, $d\varphi$ denotes the corresponding differential map $d\varphi_x : T_x \mathcal{M} \to T_{\varphi(x)} \mathcal{M}'$ on the tangent spaces, "$\circ$" denotes the composition of maps, and we routinely treat a cotangent vector $v \in T_x^* \mathcal{M}$ as a map $v \in T_x \mathcal{M} \to \mathbb{R}$. In our case, we consider $\mathcal{M}$ to be products of the gauge group ($\times \mathbb{R}$), and the adjoint state can be identified with a Lie algebra element.

## 6.3   Fermionic Flow Models

Having seen flow-based models designed to sample from scalar and gauge theories, we now turn to the fermions which are unmissable elements in quantum field theories describing nature. In the context of pseudofermions, this has been explored in [116, 37].

The goal of such a model is to use the joint distribution eq. (212) for sampling bosonic and pseudofermionic degrees of freedom. One way to achieve this is to construct *joint autoregressive model* (cf. [37]). In this approach, the distribution of the pseudofermion field $\phi$ is modeled conditionally on the gauge variable, $U$, in a sequential manner. By factorizing the joint distribution as

$$p(U, \phi) = p(U)p(\phi|U), \tag{245}$$

with the following marginal and conditional distributions

$$p(U) \propto \det DD^\dagger[U] e^{-S_g(U)},$$
$$p(\phi|U) \propto \frac{e^{-S_{pf}(U, \phi, \phi^\dagger)}}{\det DD^\dagger[U]}, \tag{246}$$

we can first sample $U$ from the marginal distribution and then sample $\phi$ based on $U$. For this, two independent flow models modelling $q(U)$ and $q(\phi|U)$ are used. In more detaills, the first component approximates the marginal distribution, cf. eq. (51)

$$q(U) = r_m(z) \left| \det \frac{\partial f_m(z)}{\partial z} \right|^{-1} , \quad U = f_m(z) \tag{247}$$

and the second the conditional distribution

$$q(\phi|U) = r_c(\chi) \left| \det \frac{\partial f_c(\chi|U)}{\partial \chi} \right|^{-1} , \quad \phi = f_c(\chi, U), \tag{248}$$

where $z, \chi$, are samples from the base distribution which are transformed to produce the gauge and pseudofermion fields $U, \phi$, while $f_m(z)$, $f_c(z|U)$ are the flow models for the marginal and conditional distribution respectively. The proposed configuration $\{U, \phi\}$ is distributed according to $q(\phi, U) = q(U)q(\phi, U)$. For $f_m(U)$ one can use gauge-equivariant layers, but for $f_c(\phi|U)$ one needs to develop a new architecture which will map an uncorrelated Gaussian distribution

$$r(\chi) = \mathcal{N}(0, \mathbb{1}) \xrightarrow{f_c(\chi|U)} q(\phi|U) \sim e^{-\phi^\dagger (\tilde{D}_f \tilde{D}_f^\dagger[U])^{-1}\phi} \, . \tag{249}$$

When $\tilde{D} = D$ and when the flow is perfect, we have $q(\phi|U) = p(\phi|U)$. A construction of such a gauge-equivariant transformation is given in [116]. As such, this approach relies on being able to compute the inverse of the Wilson-Dirac operator. The preconditioning techniques mentioned in §5.4.1 are therefore relevant in this approach.

# 7   Outlook

There is a lot more beyond the scope of this lecture note. First, in this lecture note we only cover a few physical theories as examples. When the field theory of interest involves more types of matter fields, transforming in different representations under the spacetime symmetries, global symmetries, and gauge symmetries, the normalizing flow must be expanded to accommodate these fields and their symmetries, in a way analogous to what we discussed in this lecture. Second, variations of normalizing flow architectures such as stochastic normalizing flows [127, 128, 129] and multiscale normalizing flows [130, 131, 132] have been exploited to further address issues such as topological freezing and critical slowing down in the LFT context. Third, apart from simulating lattice configurations directly, normalizing flows have been applied in a variety of ways in the context of lattice field theories [36, 121, 122, 133, 43, 134, 71]. Moreover, there are many more generative AI methods beyond normalizing flows that have been explored in the context of lattice field theories. See for instance [135, 136, 137, 138] for some recent examples.

Looking ahead, we believe that lattice computations are an interesting challenge for AI. Its success will have far-reaching impact in various domains in physics, including particle physics, theoretical high energy physics, condensed matter physics, and more, due to the ubiquity of lattice methods in these fields. As a result, having a sampling and computational framework that avoids the prohibitive scaling of costs of traditional methods holds the promise of transforming how physicists approach their problems. Though it is by no means an easy task, partially due to the maturity of lattice QCD computation pipelines that sets a very high bar as benchmark, ongoing leaps in the field of AI suggests that the incorporation of AI methods in lattice computations is an effort that will pay off and a task that the physics community should undertake.

# A   Numerical Integration on Lie Groups

In §6.2.3, we encounter differential equations of the form (233). To numerically integrate such equations, specialized methods like the Lie-Euler method, the Crouch-Grossmann Runge-Kutta schemes, and the Munthe-Kaas Runge-Kutta methods have been developed. These techniques are designed to respect the underlying Lie group structure, ensuring that the numerical solutions also stay within the Lie group [139]. Below we briefly review two of such methods.

**Lie-Euler method**   This method adapts the classical Euler method to respect the structure of Lie groups, by setting

$$U_{n+1} = \exp\left(hZ(U_n)\right)U_n\,, \tag{250}$$

which coincides up to $\mathcal{O}(h^2)$ with the standard Euler method

$$U_{n+1} = U_n + h\dot{U}_n = U_n + hZ(U_n)U_n = (\mathbb{1} + hZ(U_n))U_n\,. \tag{251}$$

**Crouch-Grossmann Runge Kutta**   The Crouch-Grossmann [140] are methods generalizing Runge-Kutta methods to the case of Lie groups, by replacing the standard update step with a Lie-

Euler step, thereby ensuring that the numerical integration respects the structure of the underlying Lie group. Recall the $s$-stage Runge-Kutta method to integrate $\dot{z} = f(z)$, with $f : \mathbb{R}^n \to \mathbb{R}^n$, is given by:

$$z_{n+1} = z_n + h \sum_{i=1}^{s} b_i f(z_i^{(n)}),$$

and $z_i^{(n)}$ are the internal stages computed similarly with:

$$z_i^{(n)} = z_n + h \sum_{j=1}^{i-1} a_{i,j} f(z_j^{(n)}),$$

where $h$ is the time step size. In the above, the coefficients $b_i$, $a_{i,j}$ and $c_i = \sum_j a_{i,j}$ are coefficients from the Butcher tableau. For the above to have the correct expansion, they must fulfill the conditions such as

$$p = 1 : \qquad \sum_i b_i = 1 \tag{252}$$

$$p = 2 : \qquad \sum_i b_i c_i = \frac{1}{2} \tag{253}$$

$$p = 3 : \qquad \sum_i b_i c_i^2 = \frac{1}{3}, \quad \sum_{ij} b_i a_{ij} c_j = \frac{1}{6}, \quad \underbrace{\sum_i b_i^2 c_i 2 \sum_{i<j} b_i c_i b_j = 1/3}_{\text{extra condition}}. \tag{254}$$

A Runge-Kutta method has order $p$ if the coefficients $b_i$, $a_{ij}$, $c_i = \sum_j a_{ij}$ fulfill the following order conditions up to order $p$ (without the extra condition).

Generalizing this to Lie groups, an $s$-stage Crouch-Grossmann method is given by

$$U_{n+1} = \exp\left(h b_s Z_s^{(n)}\right) \cdot \cdots \cdot \exp\left(h b_1 Z_1^{(n)}\right) U_n,$$

where $Z_i^{(n)} = Z(U_i^{(n)}) \in \mathfrak{g}$ is the Lie algebra-valued vector field at the internal stage $i$ that generally depends on $U_i^{(n)}$. The internal stages $U_i^{(n)} \in G$ are computed using:

$$U_i^{(n)} = \exp\left(h a_{i,i-1} Z_{i-1}^{(n)}\right) \cdot \cdots \cdot \exp\left(h a_{i,1} Z_1^{(n)}\right) U_n.$$

When the vector field $Z$ has explicit time-dependence, we note that in evaluating $Z_i^{(n)}$, we set the time to be $t_i^{(n)} = t_n + c_i h$. When the Lie group is non-Abelian, the Runge-Kutta coefficients must satisfy extra conditions for $p \geq 3$, as shown in (254) [141].

## Acknowledgments

MC would like to thank Mathis Gerdes, Pim de Haan, Roberto Bondesan and Kim Nicoli for their valuable collaborations, and the organizers of the IAIFI summer school and the MLMTP23 School for the invitation to deliver this set of lectures. The work of MC is supported by the Vidi grant (number 016.Vidi.189.182) from the Dutch Research Council (NWO), the Investigator Grant from the National Science Academy of Taiwan, and the Taiwan National Science and Technology Council.

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
