# Peer review of "Lecture Notes on Normalizing Flows for Lattice Quantum Field Theories"

_SciPost Physics Lecture Notes_

## Round 1 · Referee Report · Anonymous (Referee 1) · 2025-7-27

Strengths

Fairly detailed and self-contained lecture notes on lattice field theory (LFT) and normalising flows (NFs).

Weaknesses

Remains rather formal, with no practical demonstration. Balance between LFT (2/3) and NF (1/3) could have leaning more towards NFs.

Report

The manuscript is a set of lecture notes, based on lectures given by the first author in recent years. The material concerns lattice field theory (LFT), with quite an extensive introduction to a broad range of topics, and normalising flow (NF), a collection of machine learning methods which have been applied to LFT over the past 6 years or so.

The balance is slightly off: the intro to LFT covers about 40 pages, of which most of the material can be found in excellent textbooks (e.g. Gattringer & Lang is a popular one), whereas NF covers only 20 pages, even though it is the main topic of the notes. Personally, I would have liked to have seen a more focussed presentation on NFs.

I will now comment on the NF sections: section 3 and 6. The LFT sections are fine and easy to follow for someone with a background in LFT.

Overall I suggest acceptance, after the minor comments below have been addressed.

Requested changes

Section 3.

This section introduces NF. Overall, it provides a useful and enlightening overview. I have a few comments:

Sec 3.1

1) the eq below eq 50: it is not clear whether the normalisation, i.e. the partition function Z, is to be included in the approximate equality between q() and p(). It is important to make this explicit, given the usual intractability to compute Z.

Sec 3.2

2) eq 51 is a bit unclear: phi appears on the LHS and z on the RHS, without any relation between them. In fig 1 and further down, this is clarified as: phi=f(z(0)), or phi=z(T), or phi=f(z). It would be good to make this precise early on, with unambiguous notation. (Eqs 53-55 help to clarify this.)

3) from eq 52 we may deduce that Z, see comment 1), is not necessarily matched, since only log p and log q are matched, up to a constant. Can this be clarified?

4) eq 57: using this KL divergence requires sampling from the target distribution, which is the problem we are trying to resolve. Although this is mentioned further down, I would raise this already here as a possible concern.

5) figs 3,4,5,6 (and maybe more) are not referred to in the text. Please refer to all figures and explain what is in them.

6) fig 3: it is clear what mode collapse is and why it occurs for the reverse KL. Why the forward KL leads to the poor representation as in fig 3 is not so clear. Can this be explained better, or is fig 3 exaggerated?

7) figs 4, 5, 14 all show a building block of a transformation. But in figs 4, 14 'time' goes down and in fig 5 'time' goes up. Also the choices of boxes and actions is not the same. It would be helpful, especially in lecture notes (!), if a uniform graphical presentation is followed.

8) fig 4 and the eqs next to it: I can see that the RHS is supposed to be the inverse of the LHS. But what about s_b and t_b? The eqs suggest that s_b(z_a) on the LHS and s_b(phi_a) are the inverses, and the same for t_b. Is this meant to be the case, or is the notation a bit sloppy?

9) below eq 62: active and passive sub lattices. This is common in LFT with checkerboard-style algorithms, so I am not sure whether this is worse for the NF-based algorithms. Can this be clarified?

10) eq 68: I am not sure whether I understood why this is a loss function. Is the meaning that the loss function is a function of z(t_f), whose form has not yet been given?

11) eq 69: it may be worth stating that a(t) is independent of theta, which is used in eq 73 (?) Or is it not, looking at eqs 70 and 71 (?). Please clarify.

12) fig 6: 'a single call of the ODE solver', but still iterated during training, right? For the generation, a single forward call is needed for each new configuration?

13) eq 81: the implementation of the symmetry leads to a specific form of weight sharing. Is this correct?

Section 6.

This section provides a useful, albeit quite formal, discussion of gauge symmetry and equivariance. It would be nice to see implementations with actual numerical experiments, but I can see that that is not in scope for these lecture notes.

14) the acronym NVP is never given in full.

Sec 6.1

15) page 52: arbitrary element U, then the process is given for W. I assume this is meant to be U?

16) eq 219: Lie algebra vs group element. Could it have been different, or am I missing the point?

17) eq 220 and the discussion: it might help to add a figure?

Sec. 6.2

18) below eq 224: 'context functions': this concept has not been introduced yet. What does it mean? They are called I, but are not the identity. A few lines down, the I's are called features. This could be presented and explained a bit better.

19) fig 14 is comparable to figs 4,5, see comment 7. But the starting point is U_mu(x), which is somewhere on the RHS, quite hidden. I would expect that quantity to appear at the top of the graph, for clarity.

20) fig 15: for colour-blind people (like myself) the distinction between green and orange was challenging. Hence it took me longer than needed to realise one is really updating only the single vertical link. This then helped in understanding the difference between passive and frozen. I think fig 15 can be improved by putting this single link in the centre, and not the green plaquette.

21) i-ResNet has not been introduced in these terms (what is i?)

21) trivialising flow: 'first, he noted...'. We all know who 'he' is, but it is better to put the name.

22) eq 242: linear dependence --> nonlinear dependence.

Sec. 6.3

23) eq 246: p(U) depends on det DD^(U). To sample from p(U), how is the det incorporated? Explicit evaluations of the det are expensive. Below it is suggested that the flow model for the marginal distribution f_m(z) is easy, but this needs DD^(U). Please clarify.

Section 7

While I share the enthusiasm with the author to explore these methods, the sentence 'prohibitive scaling of costs of traditional methods' is too strong, given that a) current LFT simulations take place at the physical point, taking the continuum limit; b) the scaling of NFs is far from settled and might be prohibitively expensive (!) So I would suggest to weaken this statement somewhat.

Recommendation

Ask for minor revision

---

## Round 1 · Referee Report · Anonymous (Referee 2) · 2025-8-14

Strengths

Concise and easy-to-read presentation of a timely topic (normalizing flows) with focus on its application in an important field (lattice field theories).

Weaknesses

Not well balanced with too much lattice field theory details given on aspects which are not really relevant for the main topic of the lecture.

Report

The lecture notes cover the application of normalizing flows (NFs) to lattice field theories (LFTs) in a concise and easy-to-read way. The focus is mainly on lattice gauge theory (LGT) which is probably the most interesting application. For students not familiar with LFTs and LGT the notes provide a compact introduction to the field which should allow them to understand the specifics of the NF when applied to LFTs and LGT. On the other hand, for students not familiar with NFs, the notes provide the necessary motivation and a good overview of the basics of the NF techniques with the necessary references.

The notes are not so well balanced: the discussion of fermions on the lattice is too extensive, with too much details given that are not relevant for the fermionic NF, e.g., chiral properties. In contrast, the part on the fermionic normalizing flow is very minimal and sketchy.

The background on lattice field theory is also not well balanced. For example, a lot of space and technical details are given to the topic of improved gauge actions and different smearing schemes which are not really relevant for the rest of the lectures.

Nevertheless, I recommend publication of the lecture notes once the minor comments and corrections listed below are taken into account.

Requested changes

Table on pg. 5 is not referenced from the text. It contains refernce to the gauge field which has not been introduced at tht point.

Figure on pg.6 is not referenced in the text.

Footnote 2 should appear after the full stop, not before.

Pg.10, 6th line: "a topological observables A" $\rightarrow$ "a topological observable A"

Pg.12, 4th line after Eq.(33): "The corresponding classical solution is" $\rightarrow$ "The corresponding classical solutions are"

Section 3.1, 3rd line: Ref. [35] is repeated

Pg.15, 1st line of last paragraph: "by a set trainable parameters" $\rightarrow$ "by a set of trainable parameters"

Pg.18, 2nd paragraph: "This property makes reverse KL is a convenient choice" $\rightarrow$ "This property makes reverse KL a convenient choice"

Figure 3 on pg.18 is not referenced in the text.

Section 3.3, 1st sentence: why "Another"? There were no other families mentioned.

Figure 4 on pg.19 is not referenced in the text.

Pg.22, 2nd-to-last paragraph, 1st sentence: "Neural ODEs" $\rightarrow$ "neural ODEs"

Eq.(83): remove the full stop

Before eq.(93): "with $\hat \mu$ denotes" $\rightarrow$ "with $\hat \mu$ denoting"

Pg.30, 2nd-to-last line: the parameters $\omega_{\mu\nu}$ are not defined.

Pg.34, figure 12 is not referenced in the text. In addition, the figure is not compatible with eq.(128) which it should probably illustrate?

Pg.40, after eq.(40): "vanishes at the continuum limit" $\rightarrow$ "vanishes in the continuum limit"

Three lines below: "in the continuous limit" $\rightarrow$ "in the continuum limit"

Pg.45, last line: remove the "and", otherwise this is not a complete sentence.

Section 5.3.2, 2nd-to-last sentence sounds strange

Pg.49, 1st line: "to be doubler-free Dirac-Wilson operator" $\rightarrow$ "to be the doubler-free Dirac-Wilson operator"

Pg.49, footnote 9: "satisfy" $\rightarrow$ "satisfies"

Pg.50: "estimattion" $\rightarrow$ "estimation"

Pg.50, after eq.(210): repetition of "which"

Pg.53, line above eq.(220): "transofmration" $\rightarrow$ "transformation"

Section 6.2.1: the acronym "NVP" is not intoduced

Pg.55, section 6.2.2, 3rd line of 3rd paragraph: repetition of "both"

Pg.59: 2nd line below eq.(246): "detaills" $\rightarrow$ "details"

Refs.[59] and [65] are identical

Refs.[131] and [133] are identical

Refs.[122] and [134] are identical

Recommendation

Ask for minor revision

---

## Editorial Decision

resubmitted